# Adoption of Deep-Learning Models for Managing Threat in API Calls with Transparency Obligation Practice for Overall Resilience

**DOI:** 10.3390/s24154859

**Published:** 2024-07-26

**Authors:** Nihala Basheer, Shareeful Islam, Mohammed K. S. Alwaheidi, Spyridon Papastergiou

**Affiliations:** 1School of Computing and Information Science, Anglia Ruskin University, Cambridge CB1 1PT, UK; shareeful.islam@aru.ac.uk; 2Research and Innovation, MAGGIOLI S.P.A., Andrea Papandreou 19 Marousi, 151 24 Athens, Greece; spyros.papastergiou@maggioli.gr; 3Cybersecurity Consultancy Services Department, Securology, Jeddah 23334, Saudi Arabia; mohammed@securology.net; 4Department of Informatics, University of Piraeus, 185 34 Pireas, Greece

**Keywords:** deep learning, SHAP, transparency obligation, API security, threat management, control, vulnerability

## Abstract

System-to-system communication via Application Programming Interfaces (APIs) plays a pivotal role in the seamless interaction among software applications and systems for efficient and automated service delivery. APIs facilitate the exchange of data and functionalities across diverse platforms, enhancing operational efficiency and user experience. However, this also introduces potential vulnerabilities that attackers can exploit to compromise system security, highlighting the importance of identifying and mitigating associated security risks. By examining the weaknesses inherent in these APIs using security open-intelligence catalogues like CWE and CAPEC and implementing controls from NIST SP 800-53, organizations can significantly enhance their security posture, safeguarding their data and systems against potential threats. However, this task is challenging due to evolving threats and vulnerabilities. Additionally, it is challenging to analyse threats given the large volume of traffic generated from API calls. This work contributes to tackling this challenge and makes a novel contribution to managing threats within system-to-system communication through API calls. It introduces an integrated architecture that combines deep-learning models, i.e., ANN and MLP, for effective threat detection from large API call datasets. The identified threats are analysed to determine suitable mitigations for improving overall resilience. Furthermore, this work introduces transparency obligation practices for the entire AI life cycle, from dataset preprocessing to model performance evaluation, including data and methodological transparency and SHapley Additive exPlanations (SHAP) analysis, so that AI models are understandable by all user groups. The proposed methodology was validated through an experiment using the Windows PE Malware API dataset, achieving an average detection accuracy of 88%. The outcomes from the experiments are summarized to provide a list of key features, such as FindResourceExA and NtClose, which are linked with potential weaknesses and related threats, in order to identify accurate control actions to manage the threats.

## 1. Introduction

System-to-system communication via Application Programming Interfaces (APIs) has now been widely adopted to enable various software application programs to effectively communicate with each other. Whether it is applications within the same organization or those handled by external parties, APIs play a crucial role in making sure information can be exchanged effectively. Recent data shows a big surge in API-based communication: growing at a rate of about 32.5% every year from 2021 to 2028, it is worth a whopping USD 3034 million [1]. Hence, API-driven communication has now been adopted across various sectors due to the significant benefits relating to smooth integration among applications, enabling scalability in a cost-effective way, improved operational efficiency, and many more benefits [2]. However, as the trend towards API-driven communication continues to rise, it brings a new set of threats and related risks. Cybercriminals are now focusing more on targeting API calls to exploit weaknesses for executing various threats relating to service disruption and the exposure of sensitive data [OWASP]. API security breaches have accelerated by 400% over the past few years, with 90% of businesses reporting at least one security issue linked to APIs [3]. There is an indispensable need to tackle this threat for safe and secure API-drive communication. However, this task is challenging due to the evolving threat context targeting API calls and the large volumes of traffic associated with system-to-system communication. In this context, AI-based models such as deep learning are widely considered to understand API data and extract threats related to the API call. Several existing works have developed AI models that use API analysis to bolster defences against malicious software and offer more nuanced and effective security solutions [4,5,6]. Such works focus on distinguishing malware from benign software based on API data sets with different accuracy levels. However, these works mainly focus on the accuracy of the model, and the interpretation of the model outcome is overlooked. Additionally, AI-enabled systems, depending on the context, are now required to disclose certain information about the entire lifecycle of the AI model to meet the transparency obligations as proposed by the EU AI Act [7,8]. This would require domain-specific knowledge from the developers or suppliers in order to offer explanations and related information about the AI model and the quality of the datasets to individuals and related user groups [8]. This task is challenging due to the varying types of datasets, the explainability of the AI models, and their lifecycle. However, the practice of transparency obligations is one of the key priorities of the EU AI Act to ensure that AI-enabled application providers offer the relevant user detailed information about the working mechanism, data use, and decision-making process of the AI model [8].

Within this context, this paper contributes to identifying and managing threats in API-driven communication. The work adopts deep-learning models for this purpose and uses Explainable AI (XAI) to extract key features related to API threats. The integration of SHapley Additive exPlanations (SHAP) allows the models to further enhance the explainability of the results and subsequently provides a deep understanding of prominent features influencing threat recognition [9]. It also considers several dimensions for transparency-obligation practices. The paper offers three major contributions. Firstly, it makes use of deep-learning models, i.e., MLP and ANN, for threat detection based on API calls. These models can infer patterns from a large API dataset, enhancing the accuracy of the threat detection process. The identified threats are analysed with common weakness patterns using the open-source security intelligence of MITRE CWE to identify accurate controls to tackle the threats. Secondly, transparency obligation is uniquely considered from four distinct dimensions: data, methodology, model outcome, and model explainability. This allows for systematically explaining and understanding the AI models and their evaluation to all user groups. Finally, the applicability of the proposed integrated architecture is demonstrated through an experiment using the Windows PE Malware API dataset [10] to identify and manage threats by mapping them with open intelligence weaknesses and NIST 800-53 controls [11]. The result of the experiment achieved an average accuracy of 88% for the deep-learning models. Such accuracy justifies that our models function well, thus making API-enabled systems more secure and strengthening the security framework of these systems. Furthermore, transparency obligations were strictly adhered to, ensuring that all processes were adequately documented and easily accessible. This includes detailed documentation of the dataset, preprocessing processes applied, model training, evaluation metrics, and the rationale for feature selection using the SHAP framework. Additionally, it is necessary for the AI systems developed to comply with the necessary regulations and maintain a minimum level of transparency. This is to ensure the safe and secure utilization of the systems and to increase user trust.

## 2. Related Works

There are several existing works that focus on AI-enabled API threat detection and identification. An overview of the existing works related to our proposed approach is provided in this section.

### 2.1. Threat Analysis and Management

Threat analysis is essential for developing effective threat-management strategies, as it involves identifying and assessing potential threats within a system. This process considers various data types and weaknesses to understand the associated risks and vulnerabilities [12]. Conversely, threat management focuses on addressing these threats by analysing and mitigating them at different levels, including applications, architecture, and devices. This is achieved through iterative methodologies designed to enhance existing security measures [13]. Furthermore, threats can arise due to vulnerabilities or weaknesses in the system that attackers could exploit. Understanding the difference between weaknesses and vulnerabilities is crucial. Weaknesses are general flaws or deficiencies that could potentially lead to security issues but have not yet been exploited. Vulnerabilities are specific, identifiable security flaws that can be directly exploited by threat actors [14]. However, considering weaknesses as a foundation for an effective cybersecurity strategy enables organizations to address both current and emerging threats comprehensively [12].

In the realm of cybersecurity, diverse types of threat analysis methodologies, such as risk-based, goal-based, and asset-based analyses, are employed to identify, evaluate, and prioritize potential threats. Each type of analysis offers a unique perspective and approach to understanding and mitigating threats. Risk-Based Analysis focuses on evaluating threats based on the likelihood of their occurrence and the potential impact on the organization. This approach helps prioritize risks and allocate resources effectively to address the most significant threats. Risk-based cyber threat analysis can be used to improve cybersecurity by providing valuable insights and understanding of the patterns and relationships in cyber threats faced by organizations [15]. Goal-Based Analysis, on the other hand, centres on the organization’s objectives, identifying threats that could hinder achieving these goals. This method aligns security measures with the organization’s strategic aims, ensuring that protective efforts support overall business success. This can be achieved through the use of goal recognition algorithms, which analyse attack graphs to identify the objectives of actors in a computer network [16]. Lastly, Asset-Based Analysis targets specific organizational assets, such as data, hardware, and software, assessing threats directed at these resources. This approach prioritizes the protection of critical assets, ensuring their integrity and availability. Asset-based cyber threat analysis has several advantages. It allows for the identification and evaluation of specific assets that are at risk, enabling organizations to prioritize their security measures and allocate resources effectively [17].

Furthermore, threat analysis models reveal a significant body of work dedicated to enhancing cybersecurity through systematic threat identification, analysis, and mitigation strategies. The continuous evolution of threat analysis models in cybersecurity reflects the field’s dynamic nature and the ongoing efforts to develop more effective, efficient, and context-aware strategies to counteract sophisticated cyber threats. The integration of traditional models with modern technologies and methodologies marks a significant advancement in the cybersecurity domain, offering promising avenues for future research and application in safeguarding digital assets and infrastructures.

### 2.2. API-Based Threat Detection Using Machine Learning

There are a large number of works that contribute to using machine learning techniques for API-based threat detection due to the nature and size of the data. This section presents some works that are relevant to our research. The new method by Li et al. [4], based on deep learning, discriminates between malware and benign software by learning the intrinsic features of API sequences. They achieved this outstanding performance using a proprietary encoder that captures semantic information and behavioural patterns, achieving an accuracy of 0.9731 and an F1 score of 0.9724. On the other hand, the study by Cannarile et al. [5] compared the tree-based machine learning algorithms with Recurrent Neural Networks (RNNs) for the purpose of detecting and classifying malware and found that Bi-GRU and ExtraTrees are good algorithms for detecting and classifying, respectively. RNNs have a higher recall, and CatBoost has a better F1 Score and AUC ROC. Better classification accuracy and ensemble learning techniques are recommended. Almaleh et al. [6] worked on zero-day and obfuscated malware by combining machine learning with deep-learning techniques. The model was able to achieve up to 98% accuracy using integrated logistic regression to determine the initial weights for the neural network. This, in turn, made it possible to learn the behavioural pattern of the API call sequences and thus further enhance the ability to detect. Among the most recent work in adversarial machine learning, Almousa et al. [18] proposed an active-learning approach to show that adversarial threats with exceptionally low training could minimize statistical deviations of the classifier from the target model and work well. In addition, its authors discuss the more general implications of adversarial ML for intellectual property protection and national security. In a more elaborate review, Chang et al. [19] reviewed the literature on machine and deep-learning techniques used in malware detection, focusing on the analysis of API calls in the Windows operating system. It outlines the role of feature representation from API calls, major challenges faced by the field, and projected research directions that could enhance detection capability against increasingly sophisticated threats. 

Selecting the right model for training is very important in AI because it determines the accuracy of the prediction, which is critical in managing the existing risks. A suitable model can help identify the exact essence of the data and prioritize threats and risks. In regard to the training of models for this research, both ANN and MLP were considered. This is because when an ANN model is used, it offers easier identification of the parameters and improves the general accuracy and stability [20,21]. It also offers lower computational costs when compared to other models [22]. The research by Khan [23] utilizes the ANN model for identifying harmful malware available on the internet that can cause damage to users and systems. Despite operating on a huge database, the model shows an impressive accuracy of 99.72%. Similarly, the advantages of employing MLP models in the identification of malware are significant because MLP has high accuracy in categorizing malware [24]. MLP classifiers yield comparatively better outcomes than other techniques in terms of accuracy and precision ratings, which make MLP classifiers resourceful for dealing with complex ransomware attacks [24]. Moreover, the application of MLP and other types of machine learning models is also important when analysing the most sophisticated kinds of malware and achieving the highest detection rates for enhancing the security of computer networks [25]. Yogesh and Reddy [26] and Sai et al. [27] utilized MLP models for classifying malware in various networks, which showed benefit to the prevention of phishing and the classification of URLs between legitimate and malicious, thereby improving security countermeasures against phishing and virus spread.

In other words, API call analysis plays a critical role in enhancing threat detection capabilities, given its ability to provide detailed insights into application behaviour. The study of API calls allows for the development of high-level abstraction models that can significantly improve the system’s accuracy in detecting anomalies and threats. For example, machine learning and deep-learning methods applied to sequences of API calls have been used to distinguish benign from malicious software behaviours based on learned patterns. This approach helps not only in the identification of known threats but also in the discovery of zero-day threats, which are previously unseen and for which no signature has been built.

## 3. Overview of Transparency Obligation

Organizations are often required by regulatory bodies to be transparent in their operations and provide clear information concerning their actions, decisions, and customary practices. Transparency is considered one of the key priorities of the EU AI Act, specifically to ensure that the functionalities of AI systems are transparent so that users can understand the AI system’s decision-making process [8]. Article 52.1 emphasizes transparency and aims to improve overall user trust in AI-enabled applications [28]. We considered transparency obligation from four distinct dimensions within the scope of the proposed work. The dimensions cover the entire AI life cycle, from data description, preprocessing, and performance evaluation to explanation. Figure 1 presents these dimensions along with related characteristics that ensure the assurance of each dimension under the transparency obligation. Some of these characteristics involve measuring specific parameters, while others provide a comprehensive description, depending on their nature. For instance, for a data description, it is sufficient to provide clear information about the data type, features, missing values, and the number of instances. In contrast, for model performance evaluation characteristics, we need to obtain metrics such as accuracy, precision, recall, and F1 score. The reason for considering these dimensions is the nature of this threat detection-based research and the relevant aspects of the AI models, including the description of datasets and key features, data processing techniques, model outcome generation, and explanation.

*Data*: Quality data are one of the essential elements for AI models, and a description of the data is necessary to meet transparency obligations. Hence, certain aspects of the data within the context of cybersecurity are required in detail to enhance transparency, such as a description of the dataset and the features it includes, the type of data, data-extraction mechanisms, data management, missing values, and potential biases [29], as depicted in Figure 1. The dataset can be of different types with unique characteristics, and it is essential to thoroughly analyse this data [30]. For instance, text data comprising written content like log data and source code often needs preprocessing steps such as tokenization. Time-series data includes sequential data points recorded at specific intervals, such as threat intelligence data or anomaly detection, typically analysed using log features and rolling statistics. The characteristics of these datasets include their range, distribution, and data types. In this context, data extraction plays an important role in the collection and consolidation of data from various sources. Any missing data must be highlighted, along with the percentage of the amount of missing data and the measures that had to be taken to address these missing values. Another aspect of the data dimension is the management of data bias. The potential bias in any dataset can certainly impact the model outcome, and sampling or measurement bias is necessary to exclude all types of systematic errors that can influence the given analysis and outcomes. Sampling bias may also occur when the data collected does not properly reflect the population in question, while measurement bias may also occur due to imperfections in the instruments used or procedures applied in data collection. Data management encompasses storage formats like CSV or JSON databases as well as preprocessing steps such as cleaning and transformation. This awareness allows users to make sense of the results and understand what needs to be done, bearing in mind the existence of weaknesses that are likely to affect the validity and generality of the outcomes.*Methodology*: The methodology dimension provides detailed information about the inner workings of the chosen specific AI model [31]. This includes a detailed step-by-step description of the entire process, from data preprocessing to descriptions of the models and approaches used in their development, as shown in Figure 1. The description should be elaborate and state how each model operates and the rationale for choosing it for a given task [32]. In addition, a list of settings and configurations used for building the models, such as batch size, learning rate, number of epochs, number of hidden layers, activation functions, and regularization techniques, should be provided. An explanation regarding the rationale for using certain models over others, computational efficiency, or domain-specific considerations should also be included. This will ensure that the methodologies employed are clear and reproducible.*Model Outcome*: It is necessary to provide clear and comprehensive information about the outcomes of the AI models to assess their performance and reliability. This involves reporting the outcomes obtained by the models, including detailed performance measures such as accuracy, precision, recall, F1 score, and ROC-AUC [33], as shown in Figure 1. Several performance validation parameters, such as cross-validation, holdout validation, or bootstrapping, are utilized to provide evidence of the model’s performance reliability. Additionally, this dimension also covers the inherent limitations of a specific model, including its potential pitfalls and constraints. This includes areas where the model performed poorly, instances of overfitting and underfitting, and the confidence intervals of the predictions [34]. It is important for users to understand these aspects, as they help convey the reliability and precision of the AI model, highlighting areas where caution is needed when interpreting the results.*Model Explainability*: The final dimension of transparency obligation is model explainability, which aims to ensure that the AI models used in specific contexts are well explained and understood by relevant user groups. This includes an explainability framework along with feature importance, which influences decision-making, as depicted in Figure 1. SHapley Additive exPlanations (SHAP) is commonly used for explaining individual predictions based on cooperative game theory by attributing each of the features its importance for a particular prediction [35]. The SHAP values aggregate the feature attributions to give a detailed explanation of how each feature contributes to the model predictions. Additionally, the explainability of the AI model is required for explaining how the input features make certain predictions. For this purpose, it is necessary to use clear and simple language, visual aids, and comprehensive documentation to make it understandable to all user groups, irrespective of whether they are technical or non-technical stakeholders. The practice of transparency in the decision-making process makes it feasible to build trust and ease in making AI models used for threat detection.

## 4. System Architecture

The proposed approach, as shown in Figure 2, consists of two main sequential phases: API-related threat identification and threat management. Each phase is composed of multiple steps, providing a comprehensible and structured methodology. The first phase focuses on API-based threat detection by integrating deep-learning models. This phase includes four steps, which cover data preprocessing, model outcomes, and transparency obligation practices. The benefit of utilizing deep learning in phase one lies in its ability to process vast amounts of API data with ease. Such analysis makes it easy to predict patterns, optimize resource allocation, and anticipate maintenance needs. By integrating deep-learning algorithms into API frameworks, systems can not only react to changes but also proactively manage and optimize their operations based on predictive analytics. The output from phase one yields critical features that are then analysed in the second phase of our approach.

The second phase is the threat assessment and management phase, which includes three steps: weakness mapping through features, threat mapping through features, and controls mapping through threats. The key features from the previous phase are assessed using available and widely used open-source security intelligence frameworks to classify and understand weaknesses. The identified weaknesses are then linked to their corresponding threats, from which the modes of attack and possible threat actors will be defined. Lastly, suitable control measures are determined to mitigate the identified threats. This approach promotes holistic threat-management practices essential for enhancing the resilience and security of interactions within API-driven systems.

### 4.1. Phase 1: API-Related Threat Identification 

The main purpose of this phase is to identify threats using deep-learning models in system-to-system API-related functions. These functions are evaluated using deep-learning techniques to identify the most crucial features that could pose high risks if not properly controlled. This enhances the performance of threat detection and management. The four distinguished steps of this phase are as follows: data preprocessing for clean, well-ordered input; model training for accurate, robust model derivation; SHAP analysis for transparent interpretation of the outputs; and assurance of transparency obligation practices within the scope of the research. These steps will generate models capable of characterizing critical features from a huge dataset, thus improving the accuracy and efficiency of detecting threats. The data can be obtained from the existing API community or widely used open-source datasets, ensuring comprehensive coverage and relevance to real-world scenarios.

#### 4.1.1. Step 1.1: Data Preprocessing

The first step involves cleaning the data for inaccuracies and duplicate records and converting it to meet algorithmic requirements. This is achieved using three sub-steps, detailed below:The first sub-step is the handling of missing data. The instances in the dataset with NaN (Not a Number) values are removed and filled with 0 to handle missing data. NaN is a special floating-point value defined in the IEEE floating-point standard used to represent undefined or unrepresentable values. In the context of data analysis and data frames, NaN values are used to denote missing or null values. The features are then separated from the target variable [36].To ensure the robustness of data selection [29], SMOTE is applied to the feature set and the target variable to oversample the minority class, thus balancing the class distribution. SMOTE, or Synthetic Minority Over-sampling Technique, is a very advanced statistical approach used for class imbalance by the introduction of synthetic samples instead of mere duplication of existing ones. Developed by Nitesh V. Chawla et al. in 2002 [37], this works by identifying the minority class in a dataset and then generating new examples synthetically. SMOTE selects existing instances from the minority class and finds their nearest neighbours in the feature space. It then creates new instances along the line segments, joining a selected instance and its nearest neighbours. This approach helps in creating a more general decision boundary for the minority class, thereby enhancing the classifier’s performance on minority class examples without causing significant overfitting. By augmenting the training data using synthetic samples, SMOTE helps balance the class distribution, improving the predictive performance of machine learning models on imbalanced datasets. This ensures the robustness of data selection by providing a well-balanced dataset that better represents the underlying distribution of both classes.Finally, the resampled feature set and target variable are obtained. It also includes dividing the dataset into training and test sets for responsible evaluation of model performance.

#### 4.1.2. Step 1.2: Model Training

Once the classes are balanced by step 1.1 through the data preprocessing step, the next step is the training of the chosen AI model. The training set will then serve as an input to train the models. The selected models for threat detection are supervised, meaning they are trained on labelled data where each data instance has an associated outcome. The following is a brief description of the models: *Artificial Neural Network (ANN):* The Artificial Neural Network (ANN) is a computational model based on the neural networks that exist in the human brain. Basically, it is composed of a number of interconnected nodes or neurons across the different layers—input, hidden, and output [38]. ANNs are designed to process complex patterns through learning and adjust their internal weights and biases based on the input data, making them suitable for a wide range of applications, from pattern recognition to data classification. ANN was selected based on the fact that it can model nonlinear relationships and offer flexibility. It is vital for successful differentiation between malware and benign API calls.*Multi-Layer Perceptron (MLP):* An MLP is a neural network in which the mapping between inputs and outputs is non-linear. The Multilayer Perceptron is composed of input and output layers as well as one or more hidden layers with many neurons stacked together [39]. While in the perceptron, the neuron must have an arbitrary activation function that imposes a threshold, like ReLU or sigmoid. The reason for selecting MLP is their ability to handle complex datasets. They also have the ability to learn intricate patterns from deep-learning architectures. MLP is capable of capturing many subtle nuances in API-call behaviour to pinpoint potential threats.

As stated before, the proposed approach utilizes both the ANN and MLP models due to their unique strengths and capabilities. ANN’s capability to deal with nonlinear relationships is an added advantage, while MLP effectively addresses datasets with intricate features and learns them in the process. Using both types of architecture increases the overall level of recognition and threat modelling by combining the positive aspects of each model.

#### 4.1.3. Step 1.3: Feature Selection

The third step of the first phase is to identify which key features underlie the system’s security and threat landscape to allow a comprehensive understanding of possible vulnerabilities and risks. It shows how and in what direction each feature participates in driving the target to some value. As stated above, we worked with SHAP (SHapley Additive exPlanations), which is a game-theoretic-based approach meant to explain the output of machine learning models by taking Shapley values into account. In order to compute these values, a brute-force algorithm needs to be run on all possible feature sets so as to learn the effect of using or excluding a given feature in the prediction. The method considers the impact of each feature within all possible combinations of features, thus giving a detailed and fair attribution of contribution. The result is a solid and comprehensible explanation of the extent to which each feature contributed to the prediction, making it particularly useful for complex models or ensemble methods. This is especially useful for complex models like deep neural networks or ensemble methods.

The SHAP value for a feature j in a prediction is calculated using the formula: $$\phi_{j}=\sum_{S\subseteq N/\{j\}}\frac{\left|S\right|!\left(\left|N\right|-\left|S\right|-1\right)!}{\left|N\right|!}\;\;[f\left(S\cup \left\{j\right\}\right)-f(S)]$$
where N is the set of all features, S is a subset of features without j, f(S) is the model output without feature *j*, and f(S∪{j}) is the model output with feature *j*. This formula iteratively computes the contribution of feature *j* by comparing the change in prediction with and without the feature across all possible combinations of other features. 

In threat management, SHAP plays a crucial role by enhancing the transparency and explainability of machine learning models used for detecting and analysing threats. By breaking down how each feature contributes to a prediction, SHAP helps security analysts understand why certain alerts are triggered, which aids in prioritizing and responding to threats more effectively. This insight is invaluable at the model decision level, ensuring regulatory compliance and improving the response strategy to incidents. Moreover, SHAP explanations could indicate the direction for iterative model refinement and bias mitigation in order to make these more dependable in high-stakes conditions, such as cybersecurity and fraud detection. 

#### 4.1.4. Step 1.4: Transparency Obligation

This final step of phase 1 focuses on the transparency obligation practice using the four dimensions outlined in Section 3: It starts with data transparency by providing a comprehensive description of the dataset, including features, size of the dataset, and number of total instances. This includes outlining the type, range, importance, and handling of missing data or bias in each feature. It is also important to specify the details about the management of data and its extraction to further enhance data preprocessing and provide transparency in how issues with data are resolved.Methodological transparency gives a justification for the choice of the selected deep-learning models in terms of their ability to identify complex patterns and anomalies within the dataset. This involves a step-by-step description of the procedure from data preprocessing to model evaluation, which includes information about the algorithm, settings, and configurations for the training of the model, such as the batch size, learning rate, and number of epochs. These outcomes are used by stakeholders to assess the effectiveness and reliability of the model and understand the pitfalls, uncertainties, and limitations in the outcomes of the model. The reason for choosing ANN and MLP models is their ability to capture non-linear dependencies and their flexibility in learning complex patterns. These models also have the ability to address inherent difficulties in threat identification tasks, which provides better stability and performance for the proposed approach.After establishing methodological transparency, the next step is the performance evaluation of the model. The evaluation of the model outcome is conducted using several key metrics, such as accuracy, precision, recall, and F1 score. Accuracy measures the whole precision of the model by relating the number of correct predictions to the total number of predictions. Precision checks how well the model can identify the positive instances, while recall measures the model’s capacity to find all positive samples. The F1 score is the harmonic mean of precision and recall measurements. All the above metrics assist the stakeholders in establishing the credibility of the model and its efficacy in the identification and mitigation of threats. Such detailed assessment also helps to develop confidence among the stakeholders with respect to the outputs of the model and its capacity to solve problems in the real world.Finally, the explainability of the models is increased, as mentioned in step 3, using SHAP analysis for the selection and explanation of the features. SHAP gives a detailed explanation of how each feature contributes to the predictions of the model. This documentation is done in order to increase transparency and understanding for the stakeholders, as well as to improve the trust and reproducibility of AI threat detection and management systems.

### 4.2. Phase 2: Threat Management

This phase aims to manage threats using three distinct steps: weakness mapping through features, threat mapping through features, and controls mapping through threats, as illustrated in Figure 3. The phase begins with analysing significant system-to-system API call features derived from the previous component. These features are assessed to uncover relevant weaknesses using established open-source security intelligence frameworks like MITRE CWE [40] to classify and understand weaknesses. Once weaknesses are identified, attention shifts to outlining threats linked to these weaknesses by utilizing insights from the MITRE CAPEC [41] database to map out attack routes and potential threat agents. The last step focuses on determining suitable control measures in line with NIST SP 800-53 [11] standards to mitigate identified threats. This approach promotes holistic threat management practices essential for enhancing the resilience and security of interactions within API-driven systems.

#### 4.2.1. Step 2.1: Weakness Mapping Through Features

Once the features are extracted from the AI models, this step focuses on identifying the common weaknesses linked to these features. In this context, the proposed approach considers open intelligence security-related information from existing standards and repositories, i.e., CWE and OWASP API vulnerabilities. This step justifies the features through their associated weaknesses. Each feature should be linked to categorized weaknesses of API calls, such as improper system configuration, lack of input validation checks, uncontrolled resource consumption, incorrect use of privileges, abuse, etc.

The mapping aims to identify potential weaknesses by correlating the gathered information about API calls from the AI models to areas in which the system might be exploited or compromised due to function abuse or misconfiguration. The approach utilizes the Common Weakness Enumeration (CWE) knowledge base developed by MITRE as a foundational reference for identifying weaknesses. CWE offers multiple perspectives that assist expert analysts in pinpointing relevant weaknesses based on supporting factors like software, hardware, or communication approaches, i.e., APIs. Once weaknesses are determined, they are catalogued with corresponding CWE IDs for each function. After detecting these weaknesses, the subsequent step involves identifying threats linked to each weakness.

#### 4.2.2. Step 2.2: Threat Mapping Through Weakness 

Step 1 provides a list of common weaknesses that are mapped to the features. This step focuses on identifying the threats related to the weakness. Similar to the process of mapping weaknesses to features, the research advises subject matter experts to leverage the MITRE knowledgebase’s Common Attack Pattern Enumeration and Classification (CAPEC) as a foundational resource for cataloguing threats. The linkage between CWE IDs and CAPEC IDs facilitates the alignment of threats with previously identified weaknesses, thereby enhancing the efficiency of the threat assessment process. With all potential threats and weaknesses associated with corresponding CWE and CAPEC IDs, the findings are compiled into a comprehensive threat profile knowledgebase. This knowledgebase becomes an invaluable asset for the organization, offering a well-defined overview of the threats and weaknesses that must be addressed to improve the security and resiliency of system-to-system communication. 

#### 4.2.3. Step 2.3: Control Mapping Through Threats 

The last step aims to determine suitable mitigation controls by linking the previously identified weaknesses and threats to specific security controls from the NIST SP 800-53 control list. This step involves selecting controls that mitigate the risks associated with each identified threat to system-to-system API calls, ensuring alignment with international standards. Each weakness and threat identified through CWE and CAPEC IDs is analysed in-depth to understand the underlying root causes and the mechanisms through which these weaknesses could be exploited. By leveraging the comprehensive review of the control list provided by NIST SP 800-53, security experts can methodically determine the most appropriate controls. This holistic view allows for the implementation of layered security measures that address not only the identified weaknesses but also enhance the overall resilience of the system against similar future exploits.

## 5. Experiment and Evaluation

This section presents the experiment and the results achieved through the implementation of the proposed system architecture. It focuses on two deep-learning models and the subsequent analysis of the key features of the API calls through the SHAP framework. The aim of this experiment is to: Determine threat-related key features from the API-enabled system-to-system communication-related dataset;Implement transparency obligation practices for the entire AI life cycle;Manage the identified threats with appropriate controls.

### 5.1. Phase 1: API-Related Threat Identification

#### 5.1.1. Dataset Description

The Windows PE Malware API dataset [10] is a large dataset containing information on the cybersecurity landscape collected through Windows Portable Executable (PE) files. It is unique as it provides researchers with a comprehensive overview of Application Programming Interface (API) malware and the SHA-256 hashes of the files. One of the key contributions of the dataset is the categorization of APIs used in every PE file. This is a significant aspect of gaining deep insight into the workings of the programs and their possible security implications. The Windows PE Malware API dataset [8] is a large dataset containing information on the cybersecurity landscape collected through Windows Portable Executable (PE) files. It is unique as it makes it possible for researchers to attain a comprehensive overview related to Application Programming Interface (API) malware and SHA-256 hashes of the files. One of the key contributions of the dataset is the categorization of APIs used in every PE file. This is a significant aspect of gaining deep insight into the workings of the programs and their possible security implications. For instance, the dataset includes features like ‘LdrLoadDll,’ which indicates the frequency or presence of the ‘LdrLoadDll’ function used to load dynamic-link libraries (DLLs). This function is significant in identifying malware behaviour, as malicious software often uses DLL injection for malicious purposes. Similarly, ‘NtAllocateVirtualMemory’ represents the use of the ‘NtAllocateVirtualMemory’ function, which allocates memory in a process’s virtual address space, a common manipulation by malware for injecting and executing code. Another crucial feature, ‘NtProtectVirtualMemory’, represents the ‘NtProtectVirtualMemory’ function that changes the protection on a region of committed pages in the virtual address space of a specified process. Malware may use this function to change memory protection settings to executable, allowing the injected code to run. The dataset also includes a variety of threat families, such as Trojans, ransomware, downloaders, droppers, backdoors, stealers, spyware, adware, rootkits, and worms, among others. Additionally, the dataset offers detailed records of API call sequences for each PE file, providing insights into the behavioural patterns of software that help in understanding how malicious and benign programs operate. The size and volume of this dataset make it a flexible data source for several potential tasks in data analysis and machine learning. 

#### 5.1.2. Data Preprocessing

This initial step aims to initiate the deep-learning life cycle by preprocessing the chosen data. By following the chosen dataset, we have processed the data to ensure its suitability for model training and evaluation. Figure 4 depicts the class distribution before SMOTE, where there is a noticeable imbalance between the two classes: goodware (0) and malware (1). The goodware class has 582 instances, comprising approximately 57% of the dataset, whereas the malware class has 439 instances, comprising approximately 43% of the dataset. The goodware bar towers over the malware bar, indicating that the dataset initially contains more goodware samples. 

Figure 5 shows the class distribution after applying SMOTE. The technique has been used to address the imbalance by increasing the number of samples in the malware class to match the number of samples in the goodware class. The result is that both bars are of equal height, indicating an equal number of goodware and malware instances, with each class having 582 instances. This eliminates the class imbalance and potentially improves the performance of machine learning models trained on this data.

#### 5.1.3. Model Training 

Once the data is processed, the next step is model performance evaluation, where the trained model is assessed for its accuracy based on various metrics to determine its effectiveness in predicting outcomes on unseen data. This section presents the evaluation of the model performance between the two deep-learning models, Artificial Neural Network (ANN) and Multilayer Perceptron (MLP), based on four different key metrics: accuracy, precision, recall, and F1 score, as shown in Table 1. These metrics play a very pivotal role in deciding the capability of models for classification problems. The ANN model demonstrates a commendable overall accuracy of 91%, indicating its robustness in correctly predicting outcomes across test cases. This model also shows a high precision of 93%, meaning it has a strong ability to identify positive class labels accurately. Its recall rate stands at 89%, suggesting that it successfully captures a high percentage of actual positive instances. The balanced F1 score of 91% highlights the model’s efficiency in maintaining an equilibrium between precision and recall, marking it as highly reliable. On the other hand, the MLP model shows a slightly lower accuracy of 88%. Despite this, it excels in precision at an impressive rate of 96%, the highest in the comparison, indicating that its predictions of the positive class are extremely accurate. However, it struggles relatively with a recall of 82%, pointing to some shortcomings in identifying all actual positives. The resulting F1 score of 89% still underscores a solid performance but highlights a potential area for improvement in terms of sensitivity. 

Figure 6 depicts the confusion matrix for the two chosen models. In the case of ANN, the model has a higher number of true positives (116) and true negatives (97) compared to false positives and false negatives. The lower counts of false positives (9) and false negatives (11) suggest that the model has a good balance between precision and recall. This indicates a robust performance with high accuracy in classification, being able to classify the majority of both classes correctly. 

The MLP model also performs well, with a substantial number of true positives (102) and true negatives (101). However, it has more false negatives (25) compared to the ANN model, which suggests that while it is very precise (few false positives, only 5), it is less sensitive and tends to miss more actual positive cases. The higher false-negative rate could indicate that the MLP might struggle with recall, potentially overlooking some positive instances. 

Figure 7 illustrates the ROC curves of both the models, ANN and RNN. Both models exhibit robust classification performance, with the ANN model having an AUC (Area Under the Curve) score of 0.94 and the RNN model achieving an AUC of 0.96. Both curves start from the origin and show steep slopes at the upper left of the graph, implying high TPRs compared to FPRs. It can be observed that both models perform significantly better, as indicated by their curves being well above the diagonal line. 

#### 5.1.4. Feature Selection

This final step identifies the key features using the SHAP framework. Figure 8 highlights the relative importance of various API calls in influencing the output of the model, MLP. The most significant features, ‘FindResourceExA’, ‘RegOpenKeyExW’, and ‘LdrGetProcedureAddress’, stand out with the highest mean SHAP value, suggesting they have a substantial impact on the model’s predictions. This indicates that activities related to ‘FindResourceExA’ could be crucial in identifying malicious behaviour, particularly those involving resource manipulation or unauthorized access. Conversely, the API call ‘FindFirstFileExW’ is at the bottom of the chart, indicating it has the least influence on the model’s decisions. This implies that while ‘FindFirstFileExW’ is part of the data, its impact on determining malicious activity is minimal compared to other features. Understanding these nuances helps in refining the model and focusing on the most impactful features for malware detection. 

Figure 8 also depicts the results of a SHAP analysis for the ANN model. The most influential feature in the model’s predictions is “FindResourceExA”, as indicated by its longest bar and the highest mean SHAP value (around 0.12). This suggests that this function call within the analysed system has the strongest impact on the model’s output, possibly indicating that this feature’s activity is crucial for the model’s decision process. On the other end of the spectrum, the features with the least importance in this model, as shown by the shortest bars, are “NtEnumerateValueKey” and “NtMapViewOfSection”. These features have the lowest mean SHAP values on the chart, close to zero. This indicates that variations in these features have minimal impact on the model’s output, suggesting that their activity or behaviour is less critical for the decisions made by this model. 

Table 2 presents the overlapping features from the two models that have a significant impact on the output. At this stage, it is necessary to map each feature to the corresponding hash value (SHA) and threat category. Therefore, each entry in Table 2 represents the specific feature with its related hash and possible threat category, such as Trojan, ransomware, Downloader, and Potentially Unwanted Applications (PUA). Using this mapping, mitigation features can be proactively considered by scanning the frequency of the threats generated from the API calls. For example, a high proportion of “FindResourceExA” appears in several categories, such as Trojans or ransomware. This indicates, based on SHAP values, that it is an important feature influencing model predictions in a security context. 

#### 5.1.5. Transparency Obligations 

Transparency obligation in this context, as stated in Section 4, considers four dimensions to provide a clear and understandable AI model to the stakeholders, which can foster trust and accountability. 

*Data:* A dataset description gives the stakeholders information about the replicability and relevance of the data sources at a higher level and whether the dataset used in developing the model is relevant. Section 5.1.1 provides generalized descriptive information about the dataset by listing the possible types of features in the dataset, their possible distribution in the dataset, and the possible presence of biases. The dataset contains a total of 1022 instances, with 582 instances labelled as malware and 440 instances labelled as benign. It contains an impressive 271 features related to API calls, such as ‘LdrUnloadDll’, ‘WriteProcessMemory’, and ‘NtSetContextThread’. A good dataset description demystifies the black box on which the model is built and guarantees full information about the inputs that drive model predictions and decisions to the relevant stakeholders.*Methodology:* ANNs and MLPs were employed for threat detection because they can model complex patterns and handle large datasets. ANNs are particularly strong at capturing non-linear relationships, thus identifying complex patterns and anomalies within the data. MLPs also learn feature representations from raw data, which reduces the effort required for manual feature engineering and increases the chance of detecting subtle and highly sophisticated threats. They are scalable, speeding up the computation of high-dimensional and copious quantities of data, and their architecture allows for parallel processing, enabling real-time threat detection. Additionally, ANNs are highly accurate, which is a fundamental evaluative criterion for reducing false positives and false negatives in threat detection. Their self-adaptability allows them to adjust to new data, remaining effective against new types of threats. These qualities collectively make ANNs and MLPs the top models for robust and reliable threat detection.

As depicted in Table 3, we use both the ANN and MLP for threat detection because of their potential in modelling complex patterns and their capabilities for handling large datasets. The former was implemented with an architecture having three dense layers using the sigmoid activation function and the binary cross-entropy loss function to manage the binary classification task. The model was trained using the Adam optimizer, which is known for its efficiency with respect to gradient-based optimization, using a learning rate of 0.001. Training was carried out with batch sizes of 10 and an epoch limit of 100, with early stopping to prevent overfitting. 

Similarly, the MLP model comprised three dense layers, and the data relationships, which were most likely non-linear, had to be captured by the functioning of the ReLU activation function. In the same way, it used the binary cross-entropy loss function and was optimized with the Adam optimizer at a learning rate of 0.001. In the same way, the MLP model was trained over 100 epochs using the same batch size of 10 in order to avoid variations in training conditions. The use of ReLU in MLP, compared to sigmoidal activation functions in the other models, allows training quite effectively, as it eases the vanishing gradient problem that is often prevalent in deep architectures. Configurations in both models are set to portray robust processing of high-dimensional data and adaptability under dynamic threat environments. 

*Model Outcome:* We use both the ANN and MLP in threat detection because of their potential for modelling complex patterns and handling large datasets. As shown in Table 2, the ANN, with its three dense layers and sigmoid activation function, achieves a high accuracy of 91%, precision of 93%, recall of 89%, and an F1 score of 91%. The MLP, also with three dense layers and ReLU activation function, shows an accuracy of 88%, a notably high precision of 96%, a recall of 82%, and an F1 score of 89%. The ANN demonstrates a better balance between precision and recall, while the MLP excels in precision but has a lower recall. Confusion matrices and ROC curves further illustrate these performance differences, providing a comprehensive evaluation of both models.*Model Explainability:* As briefed in step 5.1.4, SHAP is especially useful for deep-learning model interpretability. It provides a clear and mathematically founded avenue to attribute the contribution of each feature towards the model’s predictions, making the decision process transparent and understandable. Within the domain of malware detection, SHAP identifies the most and least influential features in driving improvements in model development and requires a focus on the most important characteristics of the data. This is essential for the development of trustworthy and reliable AI systems, particularly in high-stakes environments. As shown in Figure 5, building visualizations like feature importance plots or summary plots to display how the model makes its decisions is essential to boosting transparency. These techniques aid in making overly complex models readable for non-technical users by providing intuitive and traceable insights into the mechanization of the model at play.

### 5.2. Phase 2: Threat Management

#### 5.2.1. Step 1: Weakness Mapping through Features

System-to-system API calls are an essential functionality that enables various operations across software applications and operating systems. However, attackers can leverage these functions to exploit them, leading to compromised systems. By understanding how attackers can abuse system-to-system API calls, organizations can implement robust defences to mitigate these threats. Below is a breakdown of how each mentioned API call can be abused: **FindResourceExA and FindResourceExW:** The FindResourceExA feature ranks the highest due to the nature of the privileges provided by the API call, which links to system resources. These functions are used to locate a resource with a specific type, name, and language in the specified module. Attackers could use these functions to locate and manipulate resources within an application for malicious purposes, such as modifying application behaviours or loading unauthorized content. Furthermore, these functions could be abused to load malicious resources or manipulate resource-loading paths, leading to the execution of unauthorized code or denial-of-service conditions by exhausting system resources.**LdrGetProcedureAddress and LdrGetDllHandle:** These functions are part of the Windows Native API and are used for resolving DLLs and retrieving addresses of exported DLL functions. Attackers can load and execute malicious code libraries within the context of another process or abuse them to locate vulnerable functions within legitimate DLLs through techniques like DLL hijacking. This can be used to bypass security mechanisms, escalate privileges, or stealthily run malicious activities within legitimate processes.**NtAllocateVirtualMemory and NtFreeVirtualMemory:** These system services are used to allocate or deallocate memory in a process’s virtual address space, where attackers could use them to allocate memory for injecting malicious code or for creating unauthorized memory spaces that can bypass security controls. Misuse of NtFreeVirtualMemory can lead to improper memory handling, potentially destabilizing the system by freeing memory that is still in use, which can cause unexpected behaviours or system crashes that serve as a distraction or aid to further exploits.**NtClose:** This API is used to close handles to system objects (file, registry key, etc.). Attackers might exploit improper handle management to maintain access to compromised resources even after exploitation. Furthermore, it can lead to handle hijacking or reuse vulnerabilities, where an attacker induces a program to close a legitimate handle and then substitutes it with a malicious resource, gaining unauthorized access or causing instability in the system through improper resource management.**RegOpenKeyExW and RegQueryValueExW:** These registry functions are essential for accessing and querying the Windows Registry, a central repository for system configuration. Malicious exploitation of these functions can enable attackers to manipulate registry keys, modify critical system settings and behaviours, escalate privileges, extract sensitive system information, disable security features, or inject malicious startup scripts. These actions could be used for further attacks or to maintain persistence in a system.

In summary, the identified functions in system-to-system API calls can be mapped to several key Common Weakness Enumerations (CWEs) that highlight their potential security implications. CWE-427 (Uncontrolled Search Path Element) illustrates risks related to resource location and manipulation via FindResourceExA and FindResourceExW. CWE-114 (Process Control) and CWE-425 (Direct Request) reveal the threats posed by LdrGetProcedureAddress and LdrGetDllHandle in executing unauthorized code and locating vulnerable functions. CWE-119 (Improper Restriction of Operations within the Bounds of a Memory Buffer) and CWE-416 (Use After Free) describe the misuse of memory management functions like NtAllocateVirtualMemory and NtFreeVirtualMemory, as well as handle-management issues with NtClose. Lastly, CWE-284 (Improper Access Control) and CWE-20 (Improper Input Validation) are linked to the manipulation of registry settings through RegOpenKeyExW and RegQueryValueExW, emphasizing the critical need for robust input validation and access control measures to prevent unauthorized system modifications and enhance overall security posture.

#### 5.2.2. Step 2: Threat Mapping through Weakness

In the analysis of system-to-system API calls, the threat assessment component relies on the identified weaknesses associated with these API calls, as outlined in Step 1. It is evident that these functions present significant security threats if these API calls are not effectively managed. This could lead to unauthorized access, data leakage, system instability, and, potentially, full-system compromise if exploited. Hence, the threat landscape is shaped by the potential of these APIs to be exploited by attackers in numerous ways: **Resource Manipulation Threats:** Functions like FindResourceExA and FindResourceExW could be exploited to manipulate system resources. This could lead to the execution of unauthorized code, potentially embedding malware within legitimate processes. The risk is high because it directly affects the integrity and availability of system resources. The identified weakness can be mapped to CAPEC-389: Content Spoofing Via Application API Manipulation and CAPEC-21: Exploitation of Trusted Identifiers.**Code Execution Threats:** API calls such as LdrGetProcedureAddress and LdrGetDllHandle can be used to perform DLL hijacking or injection attacks. These attacks can bypass security mechanisms, allowing attackers to execute arbitrary code within the security context of another process, leading to privilege escalation and persistent access. The identified weakness can be mapped to CAPEC-640: Inclusion of Code in Existing Process and CAPEC-160: Exploit Script-Based APIs.**Memory Management Threats:** NtAllocateVirtualMemory and NtFreeVirtualMemory could be misused for allocating memory for malicious payloads or destabilizing the system by freeing memory improperly. These vulnerabilities could be exploited to create buffer overflows or to inject code, posing a direct threat to the confidentiality, integrity, and availability of data and processes. The identified weakness can be mapped to CAPEC-8: Buffer Overflow in an API Call.**Handle Manipulation Threats:** Improper management of system handles through NtClose can lead to unauthorized resource access or system crashes. Attackers could exploit this vulnerability to maintain persistence or to cause denial of service as a distraction for more severe attacks. The identified weakness can be mapped to CAPEC-593: Session Hijacking Abd CAPEC-196: Session Credential Falsification through Forging.**Registry Attack Threats:** RegOpenKeyExW and RegQueryValueExW allow for the manipulation of registry keys. Malicious exploitation can alter system behaviour, disable security settings, or extract sensitive information, posing a direct threat to system security and user privacy. The identified weakness can be mapped to CAPEC-386: Application API Navigation Remapping and CAPEC-384: Application API Message Manipulation via Man-in-the-Middle.

#### 5.2.3. Step 3: Control Mapping through Threats 

Determining suitable controls is the last step in the threat-management process. Threats identified in the threat assessment need to be addressed by organizations that must implement robust mitigation strategies. These strategies should aim to minimize the attack surface, enhance detection capabilities, and strengthen overall system resilience against API-related vulnerabilities. With reference to the identified threats in Step 2, the following mitigation strategies can be employed: **Resource Access Controls:** Implement strict access control measures to prevent unauthorized resource manipulation. Use role-based access controls and monitor resource access patterns to detect and respond to unusual activities that could indicate an attack. Mapping identified threats to international controls standard NIST SP 800-53: AC-3 (Access Enforcement) and AC-6 (Least Privilege).**Secure Coding Practices:** Adopt secure coding practices to mitigate the risks associated with DLL injection and hijacking. This includes code signing and using a safer API. Mapping identified threats to international controls standard NIST SP 800-53: SA-8 (Security and Privacy Architecture) and SA-11 (Developer Security Testing and Evaluation).**Memory Management Security:** Enhance security measures around memory allocation and deallocation. Implement Address Space Layout Randomization (ASLR) and code integrity checks to make it difficult for attackers to predict or manipulate memory spaces. Mapping identified threats to international controls standard NIST SP 800-53: SC-28 (Protection of Information at Rest) and SI-16 (Memory Protection).**Handle Management Audits**: Conduct regular audits of handle usage and lifecycle management within applications. Implement handle verification checks to ensure that handles are not misused or exposed to hijacking opportunities. Mapping identified threats to international controls standard NIST SP 800-53: AU-12 (Audit Generation) and SI-7 (Software and Information Integrity).**Registry Security Enhancements:** Secure the Windows Registry by restricting access to critical registry keys and monitoring registry access and modification. Employ tools that detect and alert unauthorized registry changes, which could indicate an ongoing attack. Mapping identified threats to international controls standard NIST SP 800-53: CM-5 (Access Restrictions for Change) and SI-4 (Information System Monitoring).

## 6. Discussion

System-to-system communication through API calls is fundamentally crucial in the current digital age. However, the increase in API usage has been accompanied by a significant rise in API-related security incidents, highlighting APIs as attractive targets for sophisticated cyberattacks. As gateways for vast amounts of sensitive data, APIs are increasingly targeted by cybercriminals for data breaches, service disruptions, and many more. The proposed approach presented in this paper contributes to managing the threats from the API call. 

### 6.1. Deep-Learning Model with Explainability for API Threat Detection 

Our work adopts deep-learning models, namely ANN and MLP, combined with SHAP, to provide a robust framework for detecting and managing API-based threats. The proposed work identifies the most important contributing features of the API calls, which further map to the related hash values and threat categories. This enables us to enhance our understanding of how certain API features correlate with security threats. For instance, ‘FindResourceExA’ and ‘LdrGetProcedureAddress’ are highly influential for potential malware attacks, and such API context-specific information is valuable for Cybersecurity Operations Centres, API management platforms, and software development for overall security enhancement. 

The adoption of deep-learning models plays an important role in threat identification, specifically when dealing with large amounts of complex threat-related data. The existing contributions, such as Li et al. [3], transform API sequences for deep learning to distinguish between malware and benign applications by extracting the inherent characteristics of API sequences with an accuracy of 0.9731 and an F1 score of 0.9724. Cannarile et al. [4] compared tree-based machine learning algorithms with Recurrent Neural Networks (RNNs) for identifying and categorizing malware. They found Bi-GRU and ExtraTrees to be effective for identification and categorization, respectively, while RNNs have higher recall. Though these works are well equipped to identify threats in system-to-system API calls, they do not inherently improve the explainability of the models. Additionally, the level of feature abstraction in deep learning makes it even more difficult to explain in detail how the models are making decisions from the raw input data. In this context, the proposed approach is unique in its ability to identify potential threats with high accuracy, as well as increase the explainability of the model’s decisions. This enables users, irrespective of their knowledge level, to attain a clear and accurate understanding of how particular features contribute to threat identification.

### 6.2. Adoption of Transparency Obligation Practice

The proposed approach also advocates the transparency obligation practice throughout the entire life cycle of the AI system, from the dataset to the model performance evaluation. Hence, the practical implication of integrating transparency obligation practice into deep-learning-based threat management is that it allows for the justification of the chosen controls for managing threats and ensures that users are well-informed about how the AI models are making decisions with detailed information about the dataset. Transparency is not only a regulatory requirement but also builds trust with users when adopting AI models into cybersecurity. By continuously updating stakeholders with detailed descriptions of the datasets, threats, and controls extracted from the datasets, organizations can foster a culture of openness and accountability, which is essential for maintaining the integrity and trustworthiness of AI-enabled cybersecurity. This approach enables the continuous improvement of security measures through feedback, user engagement, and trust. High-quality training data is crucial because the accuracy and reliability of the ANN and MLP models directly depend on the comprehensiveness and integrity of the data used during the training process. If the data is biased, incomplete, or not representative of real-world scenarios, the model’s ability to accurately detect and manage API-based threats could be severely compromised. 

### 6.3. Threat Management 

The proposed work provides a comprehensive view of the vulnerabilities inherent in system-to-system API calls, the corresponding threats these vulnerabilities pose, and the robust control measures necessary to manage these threats. The initial identification of weaknesses within system-to-system API calls, such as those in FindResourceExA and FindResourceExW, exposes the depth of potential security breaches, particularly in resource manipulation and unauthorized code execution. The mapping of these vulnerabilities to CWEs, such as CWE-427 for uncontrolled search path elements, underscores the broad spectrum of attack vectors that could exploit these API weaknesses. In the threat-management phase, the threats are mapped to specific MITRE CAPEC open-source intelligence, providing a clear link between theoretical vulnerabilities and practical exploitation methods. For instance, the association of LdrGetProcedureAddress and LdrGetDllHandle with CAPEC-640 and CAPEC-160 illustrates a direct pathway through which these vulnerabilities can lead to code execution threats. This mapping aids in visualizing the threat landscape and prioritizing which vulnerabilities need immediate attention based on their exploitability and potential impact [42]. Finally, the discussion transitions smoothly into control assignment, where mitigation strategies are not merely suggested but are aligned with international standards such as NIST SP 800-53. This alignment is instrumental in demonstrating how theoretical vulnerabilities and identified threats translate into actionable and standardized controls. For example, the use of AC-3 and AC-6 for resource-access controls not only mitigates unauthorized resource manipulation but also provides a structured framework for organizations to implement these controls effectively. 

Our work discusses several weaknesses in system-to-system API calls and their associated threats, which align with recent studies that emphasize the critical nature of these threats. Similar to our findings, Zhang et al. [15] identified unchecked resource manipulation and unauthorized code execution as significant risks, mapping these weaknesses to the CWE-427 and MITRE CAPEC knowledgebases with NIST SP 800-53 standards. This highlights the importance of bridging theoretical weaknesses with practical exploitation methods and implementing standardized controls to ensure comprehensive security measures. Furthermore, the research adopts an integrated approach that aligns with open-source intelligence and standards, leveraging deep learning in conjunction with threat management to ensure the cybersecurity of system-to-system API communication. This alignment makes the comprehensive threat management methodology superior and unique compared to other related approaches focused solely on deep-learning-based threat analysis. By combining these elements, the approach provides a robust methodology for identifying, assessing, and mitigating threats, offering a more holistic and effective solution for cybersecurity threat management. However, it relies heavily on human effort, expertise, and the manual merging of procedures. Specifically, the outcome of the deep-learning model needs to understand the domain experts to link with the security knowledgebase. To overcome this challenge, future efforts should concentrate on creating automation tools and seamless integration solutions that can simplify these processes, ultimately improving effectiveness, precision, and scalability in threat management.

## 7. Conclusions

Threats in API calls are one of the major security concerns due to the widespread adoption of system-to-system communication via APIs. Addressing these threats is challenging due to the diverse threat categories and the massive volume of API-related calls. This work contributes to tackling this challenge by adopting a deep-learning-based approach to detect threats and a security knowledge-based approach to analyse and manage the identified threats. In particular, the proposed approach reveals a well-structured threat identification and management strategy, emphasizing the importance of mapping system-to-system API call functions that could be exploited through the extraction of key threat-related features using deep-learning models. In addition to attaining a high accuracy of above 85% for both ANN and MLP models in threat detection, our proposed approach can efficiently surpass other methodologies by integrating XAI methods, thus improving overall decision-making and transparency. This ensures that all user groups, regardless of their technical expertise, can understand which particular attributes or features are useful in the identification of threats. Moreover, the orientation towards the usage of open-source intelligence guarantees the possibility of successful practical implementation, thereby increasing the robustness and applicability of our approach.

Transparency obligation practice is another novel contribution of this work. By maintaining and disclosing clear, accurate, and accessible information about the datasets, the decision-making, and the performance of the models, we can certainly build trust with all user types, including end users, developers, application vendors, suppliers, and others. The transparency obligation of the entire AI lifecycle not only fulfils regulatory requirements but also fosters a culture of openness and accountability, enhancing the overall effectiveness of the threat-management process. In conclusion, while our proposed approach significantly enhances the ability to detect and manage API-based threats, we are planning to address the current limitations and enhance our methodology in the future. As part of our future work, we plan to extend the XAI with the LIME framework, which allows for the analysis of the local importance of particular instances of the data set. Hence, the combination of SHAP and LIME will provide a more detailed explanation of the local and global importance of the model outcome. We would also like to deploy the proposed methodology in different cybersecurity contexts, such as log analysis or threat intelligence, as well as in different domains, such as healthcare and supply chain, as part of our future research. This will certainly allow us to demonstrate the applicability and usability of the proposed approach. Finally, the approach relies heavily on human efforts and domain knowledge for threat identification and analysis, which we plan to tackle through an automated process as a future direction of this work.

## Figures and Tables

**Figure 1 sensors-24-04859-f001:**
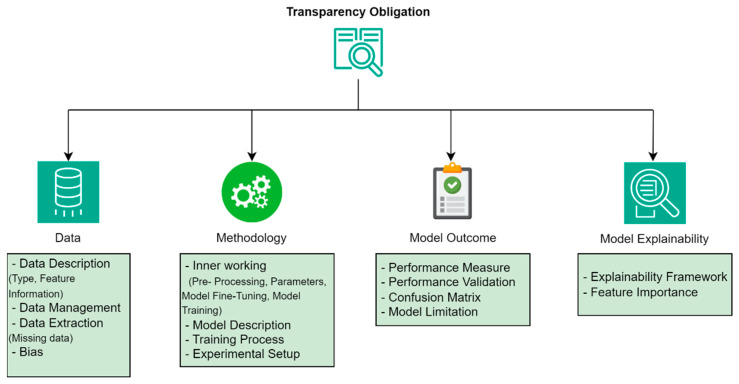
Transparency obligation dimensions.

**Figure 2 sensors-24-04859-f002:**
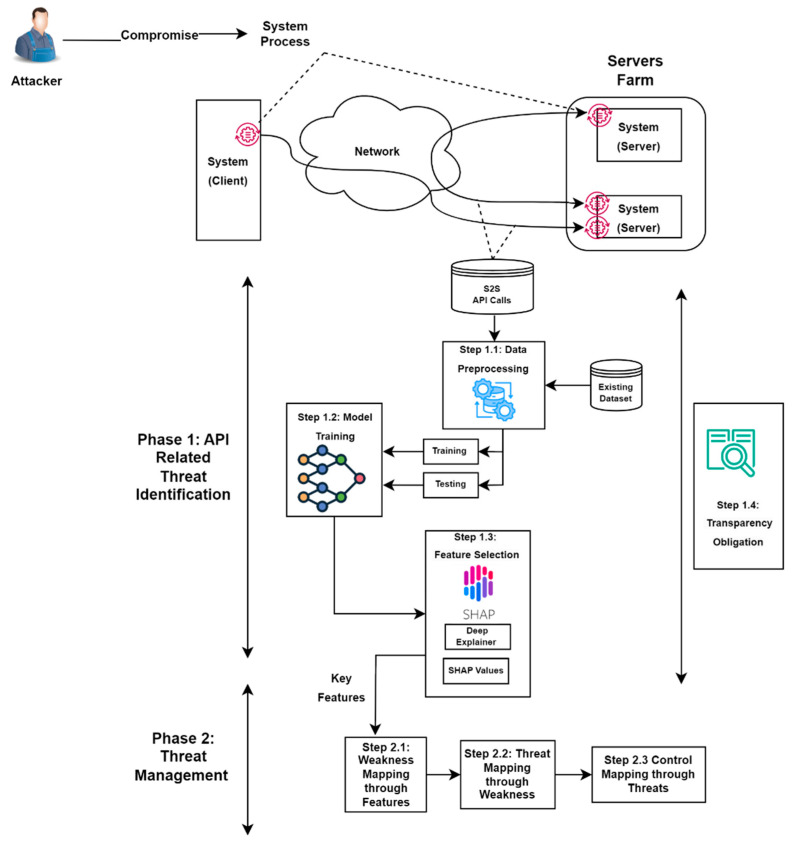
Proposed architecture.

**Figure 3 sensors-24-04859-f003:**
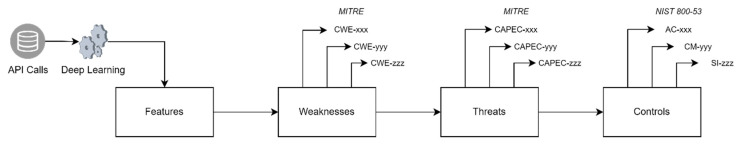
Threat management approach.

**Figure 4 sensors-24-04859-f004:**
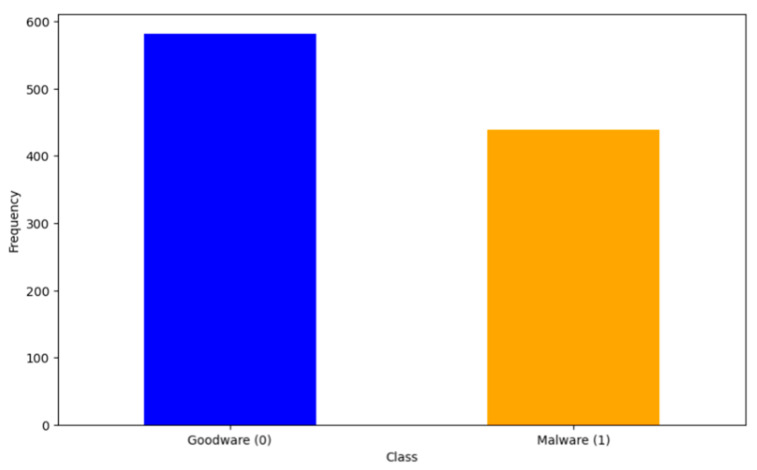
Class distribution before SMOTE.

**Figure 5 sensors-24-04859-f005:**
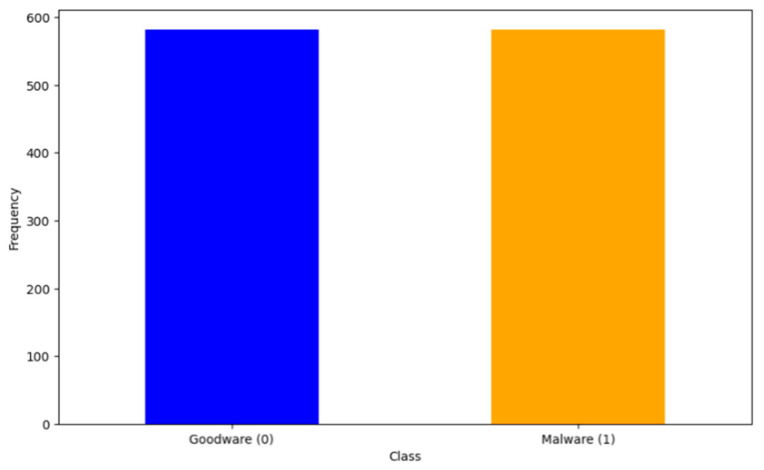
Class distribution after SMOTE.

**Figure 6 sensors-24-04859-f006:**
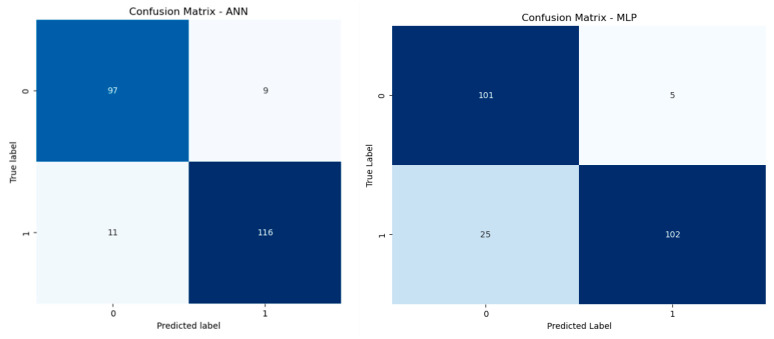
Confusion matrix of the models.

**Figure 7 sensors-24-04859-f007:**
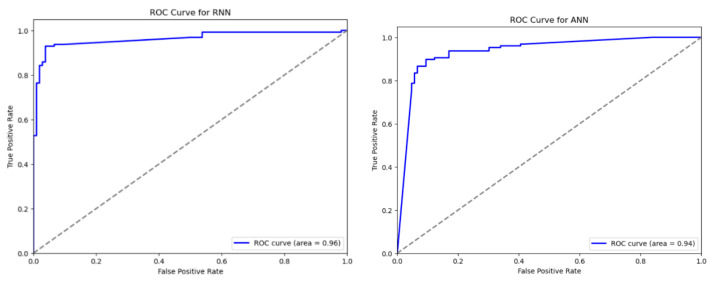
ROC curve of both the models.

**Figure 8 sensors-24-04859-f008:**
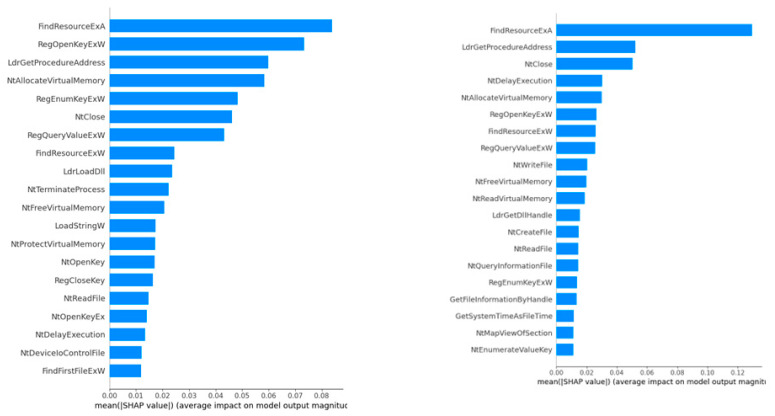
SHAP Analysis of MLP and ANN.

**Table 1 sensors-24-04859-t001:** Performance evaluation of the models.

Model	Accuracy (%)	Precision (%)	Recall (%)	F1 Score (%)
ANN	91	93	89	91
MLP	88	96	82	89

**Table 2 sensors-24-04859-t002:** Feature importance of the models.

Features	SHA	Threat Category
FindResourceExA	016101a10fa5cd4f908075fae3b9c87283122bda98fc52f1bbcc19abda7c2ffd 02d0630968348e66dd1b7f91a9784069a9f52f5a26c79976ba0efbec058788ef	Trojan, dropper, PUA, ransomware
LdrGetProcedureAddress	0808d65d091aba9c00a2d408dcebd37373d2abaf155a82991486b57cb88aeb79	Trojan
NtAllocateVirtualMemory	855f411bd0667b650c4f2fd3c9fbb4fa9209cf40b0d655fa9304dcdd956e0808	Downloader
NtClose	1125542c8a973c1f614465f0a7c7c8ebeb2dc151c5d4876bfff7cdca7fd2b15e	Ransomware
RegOpenKeyExW	992bf64436ed14bc5f5fa8d6fcf95ba658aa2b4f3e0b3d88093787bcb3b63588	Trojan
FindResourceExW	05065e5043b4b55daf9c64741eeae7ad4d7089374c92b1c41716bc3799d30d88 11ca5603b26565590265bdd0517b234dd6003c1a4b6e27d7fd3ae3b2df9b78bf 199cc9569607edeef118140ec03515ccbbd8dbc31a0f3e597fab09f6efccc2a9.	Ransomware
RegQueryValueExW	229bb814799556fa3fca643f2765adc46774ce29ec333285544b9654ccab12d9	PUA
NtFreeVirtualMemory	a3d391dac8cecf22acc946063a6c0afe18d3778cbdfc168b96b67fe5c15d9494	Trojan
LdrGetDllHandle	0a8368bab522deb622eca5805bc7bc6da0d4a6a63fae959c41c22c7d0b5ffa63	Trojan

**Table 3 sensors-24-04859-t003:** Implementation parameters of deep-learning models.

Model	Parameters	
ANN	Dense Layers	3
	Activation Function	Sigmoid
	Loss Function	Binary Cross-entropy
	Batch Size	10
	Epochs	100
	Optimizer	Adam
	Learning Rate	0.001
MLP	Dense Layers	3
	Activation Function	ReLu
	Loss Function	Binary Cross-entropy
	Batch Size	10
	Epochs	100
	Optimizer	Adam
	Learning Rate	0.001

## Data Availability

The original data presented in the reserach are openly available in [10].

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
