# Peer review of "Adoption of Deep-Learning Models for Managing Threat in API Calls with Transparency Obligation Practice for Overall Resilience"

_sensors, 2024, doi:10.3390/s24154859_

Round 1

Reviewer 1 Report

Comments and Suggestions for Authors

Dear Authors,

The paper is innovative and the results used highlight the findings of the research. However, some flaws could be improved to enhance the quality of the manuscript.

In the introduction section, the authors provided some data that lacked references to motivate or verify the statement. For example, the sentences "recent data shows a big... every year from 221 to 2028" must be referenced by some related works. Furthermore, the sentences "However, the practice of transparency... enforcement of relevant legislation" must be referenced by related works as well. Consequently, the sentences in section 2.1 (lines 118, 119, and 120) lack clarity, as the authors describe the methodologies for analyzing and managing the data, yet these methodologies are not evident in the text. Section 2.2 is pertinent in demonstrating the superiority of the methods elucidated by the authors (Li method, Cannaralimì method, and RNN) over traditional methods. Furthermore, the authors presented the ANN and MLP results in the results section, yet these methods were not mentioned in the Related Works section. To justify the exclusion or inclusion of these methods, more motivation and details are needed.

The section 3 lacked some fundamental information and references to justify the steps of the transparency obligation. For instance, the data explanation was insufficient. To provide a more comprehensive understanding, more information about the type of data, characteristics, extraction, and management of the data before the analysis is necessary. Furthermore, Figure 1 is not utilized in an appropriate manner. It is recommended that the authors utilize the figure to illustrate the phases of the transparency obligation.

The figure 2 describes the proposed methodology but in the figure is not clear the steps to develop the methodology. It is advisable to redefine the figure to explain the methodology and recall the part of the figure when the authors explain he methodology.

The authors do not provide sufficient detail regarding the data pre-processing performed in step 1 of section 4 (system architecture). The authors could enhance this section by consulting the following papers, which provide guidance on strengthening the robustness of data selection: 

10.1109/ACCT.2014.100

10.3303/CET2186039

10.1016/j.jlp.2021.104608

10.1088/1757-899X/261/1/012018

The models in question (ANN and MLP) have not been elucidated or delineated in the literature review or in the description of the macro-phases of the transparency obligation. These models must be introduced and explained in the initial steps of the paper, demonstrating their superiority to other models.

Subsequent to step 4, the subsequent steps pertain to the methodology of the transparency obligation. The authors should number the steps in a different manner to enhance the readability and comprehension of the paper.

With regard to the results, it is recommended that figures 4, 5, and 7 provide the magnitude of the values displayed in the graphs. Furthermore, the test should include a reference to the numbers and proportions of each label presented in the graph, expressed as a percentage. For example, "Figure xxx illustrates... the 20% (20 malware instances out of the total of 100) and represents the aforementioned proportion..." The same logic should be applied to all figures presented in the results section. 

The discussion section should provide some practical implications, compare the results with similar methodologies that the authors mentioned in the literature review of the paper, and mention the limitations of the research. Such details are not included in the paper.

Finally, the conclusion section could include a discussion of the following aspects:

(i) It is necessary to provide a detailed account of how this novel approach is superior to the traditional approach, including a thorough analysis and discussion of the results.

(ii) Finally, it is pertinent to illustrate the constraints of the research, which can be enhanced by the prospective advancements proposed by the authors.

Comments on the Quality of English Language

Upon reviewing the manuscript, moderate editing of the English language is necessary throughout the paper to enhance readability and coherence. This includes refining sentence structures and improving overall clarity. Addressing these linguistic aspects will significantly contribute to the manuscript's quality and readability.

Author Response

Authors’ response to Reviewers’ Comments 

Sensors (ISSN 1424-8220)MDPI 

Manuscript ID sensors- 3065709  

Title: Adoption of Deep Learning Model for Managing Threat in API calls with Transparency Obligation Practice for overall resilience 

Special Issue-Cyber Physical System: Security and Resilience Challenges and Solutions 

We want to thank the reviewers for their time and effort to review our paper and for their comments. The new version of the paper addresses all the comments raised by the reviewers. The text below indicates our responses to those comments and points out areas of the paper that have been modified based on the comments. We have added both track change and without track change version of the revised paper. 

To facilitate a more straightforward reading of our responses, we have used the following format in the text below:  

  • Each comment has been highlighted in bold.  
  • After each reviewer’s comment, our corresponding reply is provided. 

Reviewer 1 comments 

Comments and Suggestions for Authors 

Comment: The paper is innovative, and the results used highlight the findings of the research. However, some flaws could be improved to enhance the quality of the manuscript. 

Response: We thank the reviewer for the feedback and comments. All the raised flaws are addressed accordingly to enhance the quality of the manuscript. Response of each comment is added below. 

Comment: In the introduction section, the authors provided some data that lacked references to motivate or verify the statement. For example, the sentences "recent data shows a big... every year from 221 to 2028" must be referenced by some related works. Furthermore, the sentences "However, the practice of transparency... enforcement of relevant legislation" must be referenced by related works as well. Consequently, the sentences in section 2.1 (lines 118, 119, and 120) lack clarity, as the authors describe the methodologies for analysing and managing the data, yet these methodologies are not evident in the text.  

Response: Thank you for these useful comments. We have addressed all of them.  

The relevant reference is added for the recent API data in the first para of the introduction. We have also added supporting statements due to the wider adoption of API-driven communication from other literature.  

By following this comment, the sentence relating to practice of transparency is now revised to make it more consistent and related reference is added. The new sentence is as follows:  However, the practice of transparency obligations is one of the key priorities of the EU AI act   to ensure that AI -enabled application providers offer the relevant user with the detailed information about the working mechanism, data use, and decision-making process of the AI model.   

We have also revised the text in the first para of related work section 2.1 to make it more align with the threat analysis and management. In general, threat analysis is essential for developing effective threat management strategies, as it involves identifying and assessing potential threats within a system taken into account various data types and weaknesses to understand the associated risks and vulnerabilities, whereas threat management focuses on addressing these threats by analyzing and mitigating them at different levels, including applications, architecture, and devices. 

Comment: Section 2.2 is pertinent in demonstrating the superiority of the methods elucidated by the authors (Li method, Cannaralimì method, and RNN) over traditional methods. Furthermore, the authors presented the ANN and MLP results in the results section, yet these methods were not mentioned in the Related Works section. To justify the exclusion or inclusion of these methods, more motivation and details are needed. 

Response: Thank you for this useful comment. We have added some relevant literature in the areas of ANN and MLP aligning to our work in the related work as a new para in section 2.2.  

Specifically, the main motivation for selecting these two models is because of its ability to offering significant advantages in computational efficiency, parameter identification, and accuracy. Also, several research have demonstrated the high effectiveness of these models in identifying malware and enhancing cybersecurity, with impressive accuracy rates and improved countermeasures against sophisticated threats. Selecting the right model for training is very important in AI because it determines the accuracy of the prediction, which is critical in managing the existing risks. A suitable model can help in identifying the exact essence of the data and prioritizing threats and risks. Regarding the training of models for this research, both ANN and MLP have been considered. This is because when an ANN model is used, it offers efficient computations, easier identification of the parameters, and improve the general accuracy and stability. It also offers lower computational costs when compared to other models. The research by Khan utilizes the ANN model for identifying harmful malware available on the internet that can cause damage to users and systems. Despite operating on a huge database, the model shows an impressive accuracy of 99.72%. Similarly, the advantages of employing MLP models in the identification of malware are significant because MLP has high accuracy in categorizing malware. MLP classifiers yield comparatively better outcomes than other techniques in terms of accuracy and precision ratings, which make MLP classifiers resourceful for dealing with complex ransomware attacks. Moreover, the application of MLP and other types of machine learning models is also important when analysing the most sophisticated kinds of malware and achieving the highest detection rates for enhancing the security of computer networks. Yogesh and Reddy and Sai et al. utilized MLP models for classifying malware in various networks, benefiting the prevention of phishing and the classification of URLs between legitimate and malicious, thereby improving security countermeasures against phishing and virus spread. 

Additional,  text is added in step 1.2 and step 1.4 of section 4.1 to justify the inclusion of ANN and MLP by the proposed approach.  

Comment: The section 3 lacked some fundamental information and references to justify the steps of the transparency obligation. For instance, the data explanation was insufficient. To provide a more comprehensive understanding, more information about the type of data, characteristics, extraction, and management of the data before the analysis is necessary. Furthermore, Figure 1 is not utilized in an appropriate manner. It is recommended that the authors utilize the figure to illustrate the phases of the transparency obligation. 

Response: By following this comment, more information is added in section 3. Fundamental information for transparency obligation is added with relevant references. We have considered transparency from four dimensions which are not sequential. The dimension that focuses on data is now elaborated with certain aspects such as description of the dataset and underlying features, data types, data extraction mechanisms and data management, missing values, and potential biasness. Sample of relevant text is also added below.  

Quality data is one of the essential elements for AI models, and a description of the data is necessary for transparency obligations. Hence, it is crucial to describe certain specific aspects of the data within the context of cybersecurity in detail to enhance transparency, such as the description of the dataset and the features it includes, type of data, data extraction mechanisms, data management, missing values, and potential biases. The dataset can come in different types with unique characteristics, and it is essential to thoroughly analyse these data. For instance, text data comprises written content like documents, social media posts, log data, and source code, often needing preprocessing steps such as tokenization. Time-series data includes sequential data points recorded at specific intervals, such as threat intelligence data or anomaly detection, typically analysed using lag features and rolling statistics. The characteristics of these datasets include their range, distribution, and data types.  

Moreover, describing how data was extracted, from which sources, and with the help of which methods, as well as how this information is managed, for example, the formats in which data is stored and how it is pre-processed, is mandatory. Any missing data has to be highlighted along with the percentage of the amount of missing data and measures that were taken to address these missing values. Another aspect of the data dimension is the management of data bias. Potential bias in any dataset can certainly impact the model outcome, and discussing sampling bias or measurement bias is necessary to exclude all types of systematic errors that can influence the given analysis and outcomes. Sampling bias may occur when the data collected does not properly reflect the population in question, while measurement bias may occur due to imperfections in the instruments used or procedures applied in data collection. Data management encompasses storage formats like CSV or JSON databases, as well as preprocessing steps such as cleaning and transformation. This awareness allows users to make sense of the results and understand what needs to be done, bearing in mind the existence of those weaknesses that are likely to affect the validity and generality of the outcomes. 

Additionally, the figure relating to transparency obligation has now been updated to align with the new content and is better utilized by the four dimensions of transparency obligation. 

Comment: The figure 2 describes the proposed methodology but in the figure is not clear the steps to develop the methodology. It is advisable to redefine the figure to explain the methodology and recall the part of the figure when the authors explain the methodology. 

Response: Figure 2 has been redrawn to improve clarity and understanding. Specifically, the figure now includes two distinct phases with clearly delineated steps within each phase for API-related threat identification and threat management. Each phase, along with its respective steps, is clearly labelled to enhance the comprehensibility of the methodology. This restructured figure allows readers to easily follow the process and understand how each component contributes to the overall threat detection and management strategy. Additionally, the text has also been revised to describe Figure 2. 

Comment:  The authors do not provide sufficient detail regarding the data pre-processing performed in step 1 of section 4 (system architecture). The authors could enhance this section by consulting the following papers, which provide guidance on strengthening the robustness of data selection:  

10.1109/ACCT.2014.100 

10.3303/CET2186039 

10.1016/j.jlp.2021.104608 

10.1088/1757-899X/261/1/012018 

Response: Thank you for providing these interesting and important papers. We have carefully reviewed these papers and revised the text by gaining insights from the provided papers specifically relating to handling of missing data and class imbalance. The new text is now added in step 1.1 of section 4.1 relating robustness of data. The text focuses on how NaN (Not a Number) values are removed or handled, in our case as one of the key steps for strengthening the robustness of data. Additionally, we have considered Synthetic Minority Over-sampling Technique (SMOTE), which generates synthetic samples for the minority class, balancing the class distribution and preventing model bias. This allows to balance the classes, hence strengthening the robustness.  Finally, text relating to the resampled feature set and target variable are added to enhance the data processing part of the proposed approach. This ensures that the model is trained on a balanced and representative dataset, improving its generalization to real-world scenarios. Moreover, by resampling the data, we can mitigate potential biases and enhance the robustness of the model's predictions, leading to more reliable and accurate threat detection outcomes. 

Comment: The models in question (ANN and MLP) have not been elucidated or delineated in the literature review or in the description of the macro-phases of the transparency obligation. These models must be introduced and explained in the initial steps of the paper, demonstrating their superiority to other models. Subsequent to step 4, the subsequent steps pertain to the methodology of the transparency obligation. The authors should number the steps in a different manner to enhance the readability and comprehension of the paper. 

Response: We have already added description about ANN and MLP in the related work section 2.2 by following the previous comment. Additionally, both ANN and MLP also discussed as a method under the transparency obligation in step 1.4 of section 4.1.  

Step 1.4 of Phase 1, under the system architecture, has been revised to enhance readability and understanding, allowing readers to easily follow and comprehend the different steps involved in the macro phases of transparency. Additionally, in the methodological transparency section, the chosen models-ANN and MLP, have been discussed along with the reasons for their selection in this research, highlighting their ability to identify complex patterns and anomalies within the dataset.  

Specifically, ANN is selected due to its ability to capture non-linear dependencies and its flexibility in learning complex patterns. This capability is essential for accurately differentiating between benign and malicious API calls, as threats often exhibit subtle and complex behaviours that linear models may fail to detect. MLP is chosen because of its effectiveness in dealing with complex data and discovering intricate relationships within it. MLPs, with their multiple layers and non-linear activation functions, are well-suited to uncovering patterns that are not immediately apparent. This makes them particularly valuable in the context of cybersecurity, where malicious activities can be hidden within vast amounts of legitimate data. MLPs fully embody the ability of a model to capture all the differences in API call usage, enabling the detection of suspicious activity that might otherwise go unnoticed. In summary, these models were chosen for their ability to address inherent difficulties in threat identification tasks, providing better stability and performance for the proposed approach. By leveraging ANN and MLP, the system can more accurately and reliably detect threats, offering enhanced security measures. This dual-model strategy ensures a comprehensive analysis, covering a broader range of potential threat vectors and improving overall system robustness. The combination of ANN's adaptability and MLP's pattern recognition capabilities makes the proposed approach more effective in safeguarding against a wide array of cybersecurity threats. 

Comment:  With regard to the results, it is recommended that figures 4, 5, and 7 provide the magnitude of the values displayed in the graphs. Furthermore, the test should include a reference to the numbers and proportions of each label presented in the graph, expressed as a percentage. For example, "Figure xxx illustrates... the 20% (20 malware instances out of the total of 100) and represents the aforementioned proportion..." The same logic should be applied to all figures presented in the results section.  

Response: By following this comment, Figures relating to the experiment and results are provided with more detailed information. The figures now include labels that clearly depict the key aspects of the data and the outcomes of the analysis. For instance, Figure 4 depicts the class distribution before SMOTE, where there is a noticeable imbalance between the two classes: Goodware (0) and Malware (1). The Goodware class has 582 instances, comprising approximately 57% of the dataset, whereas the Malware class has 439 instances, comprising approximately 43% of the dataset. The Goodware bar towers over the Malware bar, indicating that the dataset initially contains more goodware samples.  

Figure 5 shows the class distribution after applying SMOTE. The technique has been used to address the imbalance by increasing the number of samples in the Malware class to match the number of samples in the Goodware class. The result is that both bars are of equal height, indicating an equal number of Goodware and Malware instances, with each class having 582 instances. his eliminates the class imbalance and potentially improves the performance of machine learning models trained on this data.  

However, please note that using percentages in Figure 7 to discuss the results of SHAP values might not provide the most accurate and insightful interpretation of the model's behaviour and feature contributions .  SHAP values represent the contribution of each feature to the prediction for individual instances, capturing both positive and negative impacts, and this complexity cannot be effectively conveyed by percentages. Percentages tend to oversimplify the detailed local contributions and the variability of feature importance across different instances, masking the magnitude and direction (positive or negative) of each feature's contribution. Additionally, SHAP values provide context-specific insights that percentages alone cannot capture, making it essential to use SHAP values directly for a nuanced and comprehensive understanding of feature contributions and model behaviour.  

Comment: The discussion section should provide some practical implications, compare the results with similar methodologies that the authors mentioned in the literature review of the paper, and mention the limitations of the research. Such details are not included in the paper. 

Response: Thank you for your insightful feedback. We appreciate your suggestions for enhancing the discussion section of the paper. We agree that including practical implications, comparing our results with similar methodologies mentioned in the literature review, and addressing the limitations of our research are crucial for a comprehensive analysis. By following this comment, we have compared our work with the existing contributions and added text relating to the practical implications of the proposed approach and limitation. 

Comparing to the existing work in Deep learning models for threat detection, existing works mainly focus on transforming API sequences using deep learning to distinguish between malware and benign applications by extracting inherent characteristics of API sequences with high accuracy and adoption of tree-based machine learning algorithms with Recurrent Neural Networks (RNNs) for identifying and categorizing malware. Though these works are well equipped in identifying threats in system-to-system API calls, they do not inherently improve the explainability of the models. Additionally, the level of feature abstraction in deep learning makes it even more difficult to explain in detail how the models are making decision from the raw input data. In this context, the proposed approach is unique to identify potential threats with high accuracy, as well as increase the explainability of the model’s decisions. This enables the users irrespective of their knowledge level attain a clear and accurate understanding of how particular features contribute to threat identification. 

The practical implication of integrating transparency obligation practice for the deep learning-based threat management is that it allows to justify the chosen controls for managing the threats and ensures that users are well-informed about how the AI models are making decision with detailed about the dataset. Transparency is not only a regulatory requirement but also builds trust with users for adopting AI models into cybersecurity. By continuously updating stakeholders with detailed descriptions of the datasets, threats, and controls extracted from the datasets, organizations can foster a culture of openness and accountability, which is essential for maintaining the integrity and trustworthiness of the adopting AI-enabled cybersecurity. 

This research adopts an integrated approach that aligns with open-source intelligence and standards, leveraging deep learning in conjunction with threat management to ensure the cybersecurity of system-to-system API communication. This alignment makes the comprehensive threat management methodology superior and unique comparing to other related approaches focused solely on deep learning-based threat analysis. By combining these elements, the approach provides a robust methodology for identifying, assessing, and mitigating threats, offering a more holistic and effective solution for cybersecurity threat management.  However, it relies heavily on human effort, expertise and the manual merging of procedures. Specifically, the outcome of the deep learning model needs to understand by the domain experts to link with the security knowledgebase. To overcome this challenge future efforts should concentrate on creating automation tools and seamless integration solutions that can simplify these processes ultimately improving effectiveness, precision and scalability, in threat management. 

Comment: Finally, the conclusion section could include a discussion of the following aspects: 

(i) It is necessary to provide a detailed account of how this novel approach is superior to the traditional approach, including a thorough analysis and discussion of the results. 

(ii) Finally, it is pertinent to illustrate the constraints of the research, which can be enhanced by the prospective advancements proposed by the authors. 

Response: Thank you for your insightful feedback. We acknowledge the importance of providing a detailed discussion in the conclusion to highlight the superiority of our novel approach over traditional methods and to illustrate the constraints of our research along with proposed future advancements.  

The novelty of this work is already added in introduction and discussion section. By following this comment, text is also added in the conclusion section. In particular, the proposed approach reveals a well-structured threat identification and management strategy, emphasizing the importance of mapping system-to-system API call function that could be exploited through the extracting of key threat related feature using deep learning models. In addition to attaining a high accuracy of above 85% for both ANN and MLP models, in threat detection, our proposed approach can efficiently surpass other methodologies by integrating XAI methods thus, improving the overall decision-making and transparency. This ensures that all user groups, regardless of their technical expertise, can understand which particular attributes or features are useful in the identification of threats. Moreover, the orientation with the usage of open-source intelligence guarantees the possibility of successful practical implementation, thereby increasing the robustness and applicability of our approach. Transparency obligation practice is another novel contribution of this work. By maintaining and disclosing clear, accurate, and accessible information about the datasets, the decision-making and performance of the models, we can certainly build trust with all uses types including end users, developers, application vendors, suppliers, and others. Transparency obligation of the entire AI lifecycle not only fulfils regulatory requirements but also fosters a culture of openness and accountability, enhancing the overall effectiveness of the threat management process. In conclusion, while our proposed approach significantly enhances the ability to detect and manage API-based threats, it is still essential to address the limitations related to the research. 

Comments on the Quality of English Language: Upon reviewing the manuscript, moderate editing of the English language is necessary throughout the paper to enhance readability and coherence. This includes refining sentence structures and improving overall clarity. Addressing these linguistic aspects will significantly contribute to the manuscript's quality and readability. 

Response: We have edited the entire paper to enhance readability and coherence, thereby improving the English. The current version features improved sentence structure and clarity. 

Reviewer 2 Report

Comments and Suggestions for Authors

This paper proposes a deep learning-based API threat detection and management framework that aims to improve the security of API systems. The framework combines ANN and MLP models for threat detection and uses SHAP techniques to explain the model decision-making process, while practicing transparency obligations from four dimensions: data, methods, model results, and model interpretability, providing an effective solution for API security. However, there are still some issues that need to be addressed. 

1. There are also some formatting issues in the article, such as in line 460 "5.1 . Component 1: API Related Threat Identification" should be changed to "5.1 Component 1: API Related Threat Identification"; At line 675, "5.2. Component 2: API Threat Assessment and Management" shouldbe changed to "5.2 Component 2: API Threat Assessment and Management".

2. Table 1 is not cited in the text, and it is suggested that a citation be added at an appropriate place to enhance the coherence of the article. In addition, the data in Table 1 should be labeled with the unit symbol “%” to clearly show the meaning of the data.

3. There are also grammatical problems in the article, such as “where attackers could use it to allocate memory...” in line 702, and “while” in line 704. NtFreeVirtualMemory misuse can lead to...". Capitalize the word at the beginning of the sentence.

4. The experimental section lacks comparison with other API threat detection methods. It is suggested to add a comparative analysis with existing studies to highlight the advantages and uniqueness of the proposed method.

Comments on the Quality of English Language

Minor editing of English language required.

Author Response

Authors’ response to Reviewers’ Comments 

Sensors (ISSN 1424-8220)MDPI 

Manuscript ID sensors- 3065709  

Title: Adoption of Deep Learning Model for Managing Threat in API calls with Transparency Obligation Practice for overall resilience 

Special Issue-Cyber Physical System: Security and Resilience Challenges and Solutions 

We want to thank the reviewers for their time and effort to review our paper and for their comments. The new version of the paper addresses all the comments raised by the reviewers. The text below indicates our responses to those comments and points out areas of the paper that have been modified based on the comments. We have added both track change and without track change version of the revised paper. 

To facilitate a more straightforward reading of our responses, we have used the following format in the text below:  

  • Each comment has been highlighted in bold.  
  • After each reviewer’s comment, our corresponding reply is provided. 

Reviewer 2 comments 

Comments and Suggestions for Authors 

Comment: This paper proposes a deep learning-based API threat detection and management framework that aims to improve the security of API systems. The framework combines ANN and MLP models for threat detection and uses SHAP techniques to explain the model decision-making process, while practicing transparency obligations from four dimensions: data, methods, model results, and model interpretability, providing an effective solution for API security. However, there are still some issues that need to be addressed.  

Response: Thank you for this insightful comment. We have addressed all the issues raised accordingly to improve the quality and clarity of the paper.  

Comment 1: There are also some formatting issues in the article, such as in line 460 "5.1 . Component 1: API Related Threat Identification" should be changed to "5.1 Component 1: API Related Threat Identification"; At line 675, "5.2. Component 2: API Threat Assessment and Management" shouldbe changed to "5.2 Component 2: API Threat Assessment and Management". 

Response: We have addressed the formatting issues as suggested. Specifically, the formatting in line 460 has been corrected to "5.1 Phase 1: API Related Threat Identification." Similarly, the formatting in line 675 has been updated to "5.2 Phase 2: API Threat Assessment and Management." These changes have been made to ensure consistency and clarity throughout the article. 

Comment 2: Table 1 is not cited in the text, and it is suggested that a citation be added at an appropriate place to enhance the coherence of the article. In addition, the data in Table 1 should be labeled with the unit symbol “%” to clearly show the meaning of the data. 

Response: Table 1 is now cited at an appropriate place in the text to enhance the coherence of the article. Additionally, the data in Table 1 has been labeled with the unit symbol “%” to clearly indicate the meaning of the data. 

Comment 3: There are also grammatical problems in the article, such as “where attackers could use it to allocate memory...” in line 702, and “while” in line 704. NtFreeVirtualMemory misuse can lead to...". Capitalize the word at the beginning of the sentence. 

Response: We appreciate your attention to detail and have made the necessary revisions to improve the clarity and correctness of the paper  

Comment 4: The experimental section lacks comparison with other API threat detection methods. It is suggested to add a comparative analysis with existing studies to highlight the advantages and uniqueness of the proposed method. 

Response: We appreciate your suggestions for enhancing the discussion section of the paper. We agree that including practical implications, comparing our results with similar methodologies mentioned in the literature review, and addressing the limitations of our research are crucial for a comprehensive analysis. To address this, we have incorporated the following changes to the discussion section along with the addition of the limitations of the research work. 

Deep learning models play an important role in the threat identification of system-to-system API calls, especially when dealing with large amounts of data and complex threats. Researchers such as Li et al.  have transformed API sequences for deep learning to distinguish between malware and benign applications by extracting inherent characteristics of API sequences, achieving an accuracy of 0.9731 and an F1 score of 0.9724 with a focus encoder that collects semantic data and behavioural trends. Cannarile et al. compared tree-based machine learning algorithms with Recurrent Neural Networks (RNNs) for identifying and categorizing malware, finding Bi-GRU and ExtraTrees to be effective for identification and categorization, respectively, while RNNs have higher recall. Despite the effectiveness of these models in identifying threats, they do not inherently improve model explainability. The high level of feature abstraction in deep learning further complicates the explanation of how decisions are made from raw input data. Our proposed approach addresses this by not only identifying potential threats with high accuracy but also increasing the explainability of model decisions, allowing users of all knowledge levels to understand how specific features contribute to threat identification. 

Our research discussed several weaknesses in system-to-system API calls and associated threats, aligning with recent studies that emphasize the critical nature of these issues. Similar to our findings, Zhang et al. identified unchecked resource manipulation and unauthorized code execution as significant risks, mapping these weaknesses to CWE-427. Our approach also links identified weaknesses to the MITRE CAPEC knowledgebase and aligns mitigation strategies with NIST SP 800-53 standards, a method adopted from Jones et al., who emphasized bridging theoretical weaknesses with practical exploitation methods and implementing standardized controls to ensure comprehensive security measures. This consensus highlights the importance of detailed threat analysis and robust control measures in enhancing security posture. 

Furthermore, our research adopts an integrated approach that aligns with open-source intelligence and standards, leveraging deep learning alongside threat management to ensure the cybersecurity of system-to-system API communication. This alignment makes our comprehensive threat management methodology superior to approaches focused solely on deep learning threat analysis. By combining these elements, we offer a robust methodology for identifying, assessing, and mitigating threats, providing a more holistic and effective solution for cybersecurity threat management. However, this method relies heavily on human labour, expertise, and the manual merging of procedures. While efficient, it necessitates significant time, effort, and expertise to harmonize deep learning results with the threat management process, potentially leading to inefficiencies and human errors. Future efforts should focus on developing automation tools and seamless integration solutions to simplify these processes, ultimately improving effectiveness, precision, and scalability in threat management. 

Comments on the Quality of English Language: Minor editing of English language required. 

Response: We have revised the whole paper to improve the quality of English. The current format of the paper improved overall English language. 

Reviewer 3 Report

Comments and Suggestions for Authors

The article proposes an approach to adoption of Deep Learning model for managing threat in API 2.

Remarks:

1. It is necessary to change the abstract, because it is presented in a narrative style. I recommend using the following words more often: presented, presented, described, considered, depicted, analyzed, compared, summarized, etc.

2. According to the results of "Confusion Matrix-ANN" and "Confusion Matrix-MLP", it is necessary to conduct ROC analysis and evaluate the quality of classification based on the areas under the ROC curves. The article contains a relevant reference to 23 literary sources.

Author Response

Authors’ response to Reviewers’ Comments 

Sensors (ISSN 1424-8220)MDPI 

Manuscript ID sensors- 3065709  

Title: Adoption of Deep Learning Model for Managing Threat in API calls with Transparency Obligation Practice for overall resilience 

Special Issue-Cyber Physical System: Security and Resilience Challenges and Solutions 

We want to thank the reviewers for their time and effort to review our paper and for their comments. The new version of the paper addresses all the comments raised by the reviewers. The text below indicates our responses to those comments and points out areas of the paper that have been modified based on the comments. We have added both track change and without track change version of the revised paper. 

To facilitate a more straightforward reading of our responses, we have used the following format in the text below:  

  • Each comment has been highlighted in bold.  
  • After each reviewer’s comment, our corresponding reply is provided. 

Reviewer 3 comments 

Comment: The article proposes an approach to adoption of Deep Learning model for managing threat in API 2. 

Response: We appreciate the reviewer's feedback and comments. We have addressed all the identified issues to improve the quality of the manuscript. Our responses to each comment are detailed below. 

Comment:  1. It is necessary to change the abstract, because it is presented in a narrative style. I recommend using the following words more often: presented, presented, described, considered, depicted, analyzed, compared, summarized, etc. 

Response: Thank you for your feedback regarding the abstract. We understand the need to use a more structured and formal style. We will revise the abstract to incorporate the recommended words such as presented, described, considered, depicted, analysed, compared, and summarized, ensuring that it aligns with the expected academic standards. This will help clearly communicate the key aspects of our work and its contributions.  

Comment : 2. According to the results of "Confusion Matrix-ANN" and "Confusion Matrix-MLP", it is necessary to conduct ROC analysis and evaluate the quality of classification based on the areas under the ROC curves. The article contains a relevant reference to 23 literary sources. 

Response: We have resolved this issue and added the finding to the paper. We have conducted the ROC analysis for both the ANN and MLP model to evaluate the quality of classification based on the areas under the ROC curves. This addition can significantly improve the quality of the paper and provide an enhanced evaluation of the model's performance. The ROC curves provide a visual representation of the true positive rate versus the false positive rate at various threshold settings, and the area under the curve (AUC) is used as a measure of the model's ability to discriminate between the positive and negative classes. 

The result of the ROC analysis shows that both the models exhibit robust classification performance with the ANN model having an AUC score of 0. 94 and the RNN model achieving an AUC of 0. 96. Both curves start from the origin and show steep slopes at the upper left of the graph, implying high TPRs compared to FPRs. It was observed that both models perform significantly better, as indicated by their curves being well above the diagonal line representing a random classifier. 

Reviewer 4 Report

Comments and Suggestions for Authors

      The work addresses an interesting topic from the point of view of security such as network services APIs. The paper makes an interesting theoretical proposal that is formally correct. It is also well written and easy to follow.

The theoretical proposal seems well formulated and resolved to me. However, in the experimental part there is something that doesn't fit me.

The architecture description (Figure 2) shows a client-server scheme, where the server's service requests are analyzed using its APIs. It is indicated that these interfaces are the ones to be analyzed. 

However, in the experimental part a set of malware data is used and the functions analyzed correspond to those used to obtain services from the operating system, that is, they are from another API than the one indicated in the conceptual model. This aspect must be clarified since the results obtained have a different scope.

Author Response

Authors’ response to Reviewers’ Comments 

Sensors (ISSN 1424-8220)MDPI 

Manuscript ID sensors- 3065709  

Title: Adoption of Deep Learning Model for Managing Threat in API calls with Transparency Obligation Practice for overall resilience 

Special Issue-Cyber Physical System: Security and Resilience Challenges and Solutions 

We want to thank the reviewers for their time and effort to review our paper and for their comments. The new version of the paper addresses all the comments raised by the reviewers. The text below indicates our responses to those comments and points out areas of the paper that have been modified based on the comments. We have added both track change and without track change version of the revised paper. 

To facilitate a more straightforward reading of our responses, we have used the following format in the text below:  

  • Each comment has been highlighted in bold.  
  • After each reviewer’s comment, our corresponding reply is provided. 

Reviewer 4 comments 

Comment: The work addresses an interesting topic from the point of view of security such as network services APIs. The paper makes an interesting theoretical proposal that is formally correct. It is also well written and easy to follow. The theoretical proposal seems well formulated and resolved to me. However, in the experimental part there is something that doesn't fit me. 

Response: Thank you for this comment. We have revised the experiment part of the paper and addressed the issue raised to improve the paper. 

Comment: The architecture description (Figure 2) shows a client-server scheme, where the server's service requests are analyzed using its APIs. It is indicated that these interfaces are the ones to be analyzed. However, in the experimental part a set of malware data is used and the functions analyzed correspond to those used to obtain services from the operating system, that is, they are from another API than the one indicated in the conceptual model. This aspect must be clarified since the results obtained have a different scope. 

Response: We appreciate the opportunity to clarify the aspects related to the architecture description in Figure 2 and the experimental data used in our study. Our architecture presents a generic indicative system-to-system API communication framework, encompassing system, service, and network interactions. API data can also be generated from this indicative system-to-system API call.  In this framework, API calls initiated by client applications are directed to a specific system for storage and processing via network APIs. 

Please note that the upper part of Figure 2 is an exemplary scenario which demonstrates API call related data that can be generated from the specific organizational context. However, we can also process the data from the existing widely used datasets if organizational data is not available for any specific reasons such as privacy. Therefore, our approach presented in Figure 2 considers both options, i.e., consolidation data from specific organizational context or using existing data set. In the experiment part, we have considered option two using existing widely used data set to demonstrate the applicability of the work. 

In summary, the proposed architecture presented in Figure 2 and experimental scope are aligned in their goal of enhancing API security through advanced threat detection techniques. The experimental focus on OS-level API calls provides a robust foundation for validating our approach, which can be extended to include a broader range of API interactions in future studies. 

Round 2

Reviewer 1 Report

Comments and Suggestions for Authors

Dear Authors,

The research presents the improvements suggested in the initial review, thereby enhancing the quality of the research and facilitating a deeper understanding of it. Nevertheless, a few additional modifications could be made prior to the final submission.

The clarity of Figure 1 is severely compromised. In section 3, the authors should explain the relationship between the figure and the transparency obligation. It is essential that the figure serves to support the text for the reader. In this instance, the situation is not at all clear.

In the conclusion, the authors must provide more details regarding future research. For example, the authors declare forth the proposition of implementing XAI techniques as a means of justifying the requirement of LIME. To demonstrate this assertion, they drew upon existing literature to illustrate the rationale behind their argument. Furthermore, it would be beneficial to include an additional line of future research to ascertain the applicability and usability of the frameworks in other domains and with other techniques.

Comments on the Quality of English Language

Upon reviewing the manuscript, minor editing of the English language is necessary throughout the paper to enhance readability and coherence.

Author Response

To  

Guest Editors 

Special Issue " Cyber Physical System: Security and Resilience Challenges and Solutions" 

Sensor, MDPI 

Re-submission of a paper entitle Adoption of Deep Learning Model for Managing Threat in API calls with Transparency Obligation Practice for overall resilience, Manuscript ID sensors- 3065709  

Dear Sir, 

Firstly, thank you for providing the second-round reviewers’ comments for our paper. We have further revised the manuscript according to the referees’ comments. Please see below the response:   

Reviewer 1 

Comments: The research presents the improvements suggested in the initial review, thereby enhancing the quality of the research and facilitating a deeper understanding of it. Nevertheless, a few additional modifications could be made prior to the final submission. 

Response: Thank you for this comment. We have revised all raised modifications. 

Comments: The clarity of Figure 1 is severely compromised. In section 3, the authors should explain the relationship between the figure and the transparency obligation. It is essential that the figure serves to support the text for the reader. In this instance, the situation is not at all clear. 

Response:  By following this comment, we have revised the text to explain the relationship between Figure 1 and Transparency obligation. 

We have considered transparency obligation from four distinct dimensions which is within the scope of the proposed work. The dimensions are focused to cover the entire AI life cycle from data description, pre-processing, performance evaluation, and explanation. Figure 1 presents these dimensions with related characteristics which are used to ensure the assurance of specific dimension under the transparency obligation.  The reason for considering these dimensions is due to the nature of this threat detection-based research and the consideration of relevant aspects of the AI models, including the description of datasets and key features, data processing techniques, model outcome generation, and explanation.  

We have also revised the Figure 1 to better align it with the four dimensions and related characteristics. Moreover, the text related to each dimension is also revised to provide clearer explanations and more precise details. For instance, we have revised the methodological transparency section to ensure it is clear what we are trying to achieve in this dimension. These updates ensure a comprehensive presentation of the framework, enhancing both its clarity and usability. 

Comments: In conclusion, the authors must provide more details regarding future research. For example, the authors declare forth the proposition of implementing XAI techniques as a means of justifying the requirement of LIME. To demonstrate this assertion, they drew upon existing literature to illustrate the rationale behind their argument. Furthermore, it would be beneficial to include an additional line of future research to ascertain the applicability and usability of the frameworks in other domains and with other techniques. 

Response: By following this comment, we have provided more details regarding future research. In particular, as future work, we plan to extend the XAI with LIME framework, which allows to analyse the local importance for the particular instance of the data set. Hence, the combination of SHAP and LIME will provide more detailed explanation about the feature from both local and global importance of the model outcome. We would also like to deploy the proposed methodology in different cybersecurity context such as log analysis or threat intelligence and different domain such as healthcare and supply chain as part of our future research. This will certainly allow to demonstrate the applicability and usability of the proposed approach. Finally, the approach relies heavily on human efforts and domain knowledge for threat identification and analysis, which we plan to tackle through an automated process as a future direction of this work.  

Comments: Upon reviewing the manuscript, minor editing of the English language is necessary throughout the paper to enhance readability and coherence. 

Response: The whole manuscript is reviewed to improve the readability and coherence. 

Reviewer 2 

Comment: The revised version has addressed most of my concerns, and I suggest acceptance. 

Comment: English should be improved. 

Response: The whole manuscript is revised to improve English. 

 The paper is now resubmitted with following three files 

  1. Revised paper with track change  
  1. Revised paper without track change 
  1. Cover letter with response to the reviewers’ comments. 

We hope the paper will be accepted for publication to this special issue.  

Sincerely Yours 

Nihala Basheer 

Reviewer 2 Report

Comments and Suggestions for Authors

The revised version has addressed most of my concerns, and I suggest an acceptance.

Comments on the Quality of English Language

English should be improved.

Author Response

(The authors gave the same response as above.)

Reviewer 4 Report

Comments and Suggestions for Authors

Now, the present version of the paper has clarificated my doubts.

Author Response

(The authors gave the same response as above.)
